# PITFALLS OF IN-DOMAIN UNCERTAINTY ESTIMATION AND ENSEMBLING IN DEEP LEARNING

**Arsenii Ashukha** [*]
Samsung AI Center Moscow, HSE[‡]
aashukha@bayesgroup.ru

**Alexander Lyzhov** [*]
Samsung AI Center Moscow, Skoltech,[†] HSE[§]
alyzhov@bayesgroup.ru

**Dmitry Molchanov** [*]
Samsung AI Center Moscow, HSE[‡]
dmolch@bayesgroup.ru

**Dmitry Vetrov**
HSE,[‡] Samsung AI Center Moscow
dvetrov@bayesgroup.ru

## ABSTRACT

Uncertainty estimation and ensembling methods go hand-in-hand. Uncertainty estimation is one of the main benchmarks for assessment of ensembling performance. At the same time, deep learning ensembles have provided state-of-the-art results in uncertainty estimation. In this work, we focus on in-domain uncertainty for image classification. We explore the standards for its quantification and point out pitfalls of existing metrics. Avoiding these pitfalls, we perform a broad study of different ensembling techniques. To provide more insight in this study, we introduce the *deep ensemble equivalent* score (DEE) and show that many sophisticated ensembling techniques are equivalent to an ensemble of only few independently trained networks in terms of test performance.

video / code / blog post

## 1 INTRODUCTION

Deep neural networks (DNNs) have become one of the most popular families of machine learning models. The predictive performance of DNNs for classification is often measured in terms of accuracy. However, DNNs have been shown to yield inaccurate and unreliable *probability* estimates, or predictive uncertainty (Guo et al., 2017). This has brought considerable attention to the problem of uncertainty estimation with deep neural networks.

There are many faces to uncertainty estimation. Different desirable uncertainty estimation properties of a model require different settings and metrics to capture them. *Out-of-domain* uncertainty of the model is measured on the data that does not follow the same distribution as the training dataset (out-of-domain data). Out-of-domain data can include images corrupted with rotations or blurring, adversarial attacks (Szegedy et al., 2013) or data points from a completely different dataset. The model is expected to be resistant to data corruptions and to be more uncertain on out-of-domain data than on in-domain data. On the contrary, *in-domain* uncertainty of the model is measured on data taken from the training data distribution, i.e. data from the same domain. In this setting the model is expected to produce reliable probability estimates, e.g. the model shouldn't be too overconfident in its wrong predictions.

**Pitfalls of metrics** We show that many common metrics of in-domain uncertainty estimation (e.g. log-likelihood, Brier score, calibration metrics, etc.) are either not comparable across different models or fail to provide a reliable ranking. We address some of the stated pitfalls and point out more reasonable evaluation schemes. For instance, although *temperature scaling* is not a standard for ensembling techniques, it is a must for a fair evaluation. With this in mind, the

---

[*]Equal contribution   [§]HSE refers to National Research University Higher School of Economics
[†]Skoltech refers to Skolkovo Institute of Science and Technology
[‡]HSE refers to Samsung-HSE Laboratory, National Research University Higher School of Economics

*calibrated log-likelihood* avoids most of the stated pitfalls and generally is a reasonable metric for in-domain uncertainty estimation task.

**Pitfalls of ensembles** Equipped with the proposed evaluation framework, we are revisiting the evaluation of ensembles of DNNs—one of the major tools for uncertainty estimation. We introduce the *deep ensemble equivalent* (DEE) score that measures the number of independently trained models that, when ensembled, achieve the same performance as the ensembling technique of interest. The DEE score allows us to compare ensembling techniques across different datasets and architectures using a unified scale. Our study shows that most of the popular ensembling techniques require averaging predictions across dozens of samples (members of an ensemble), yet are essentially equivalent to an ensemble of only few independently trained models.

**Missing part of ensembling** In our study, test-time data augmentation (TTA) turned out to be a surprisingly strong baseline for uncertainty estimation and a simple way to improve ensembles. Despite being a popular technique in large-scale classification, TTA seems to be overlooked in the community of uncertainty estimation and ensembling.

## 2 SCOPE OF THE PAPER

We use standard benchmark problems of image classification which comprise a common setting in research on learning ensembles of neural networks. There are other relevant settings where the correctness of probability estimates can be a priority, and ensembling techniques are used to improve it. These settings include, but are not limited to, regression, language modeling (Gal, 2016), image segmentation (Gustafsson et al., 2019), active learning (Settles, 2012) and reinforcement learning (Buckman et al., 2018; Chua et al., 2018).

We focus on in-domain uncertainty, as opposed to out-of-domain uncertainty. Out-of-domain uncertainty includes detection of inputs that come from a completely different domain or have been corrupted by noise or adversarial attacks. This setting has been thoroughly explored by (Ovadia et al., 2019).

We only consider methods that are trained on clean data with simple data augmentation. Some other methods use out-of-domain data (Malinin & Gales, 2018) or more elaborate data augmentation, e.g. mixup (Zhang et al., 2017) or adversarial training (Lakshminarayanan et al., 2017) to improve accuracy, robustness and uncertainty.

We use conventional training procedures. We use the stochastic gradient descent (SGD) and use batch normalization (Ioffe & Szegedy, 2015), both being the de-facto standards in modern deep learning. We refrain from using more elaborate optimization techniques including works on super-convergence (Smith & Topin, 2019) and stochastic weight averaging (SWA) (Izmailov et al., 2018). These techniques can be used to drastically accelerate training and to improve the predictive performance. Thus, we do not comment on the training time of different ensembling methods since the use of these and other more efficient training techniques would render such a comparison obsolete.

A number of related works study ways of approximating and accelerating prediction in ensembles. The distillation mechanism allows to approximate the prediction of an ensemble by a single neural network (Hinton et al., 2015; Balan et al., 2015; Tran et al., 2020), whereas fast dropout (Wang & Manning, 2013) and deterministic variational inference (Wu et al., 2018) allow to approximate the predictive distribution of specific stochastic computation graphs. We measure the raw power of ensembling techniques without these approximations.

All of the aforementioned alternative settings are orthogonal to the scope of this paper and are promising points of interest for further research.

## 3 PITFALLS OF IN-DOMAIN UNCERTAINTY ESTIMATION

No single metric measures all the desirable properties of uncertainty estimates obtained by a model of interest. Because of this, the community is using many different metrics in an attempt to capture the quality of uncertainty estimation, such as the Brier score (Brier, 1950), log-likelihood (Quinonero-Candela et al., 2005), metrics of calibration (Guo et al., 2017; Nixon et al., 2019), performance of misclassification detection (Malinin & Gales, 2018), and threshold–accuracy curves

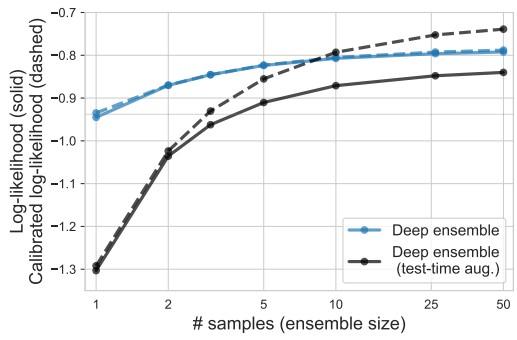 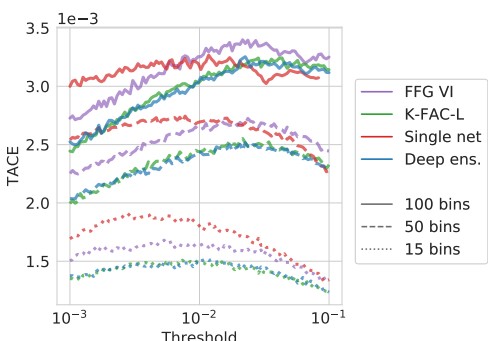

Figure 1: The average log-likelihood of two different ensembling techniques for ResNet50 on ImageNet dataset before (solid) and after (dashed) temperature scaling. Without the temperature scaling, test-time data augmentation decreases the log-likelihood of plain deep ensembles. However, when the temperature scaling is enabled, deep ensembles with test-time data augmentation outperform plain deep ensembles.

Figure 2: Thresholded adaptive calibration error (TACE) is highly sensitive to the threshold and the number of bins. It does not provide a consistent ranking of different ensembling techniques. Here TACE is reported for VGG16BN model on CIFAR-100 dataset and is evaluated at the optimal temperature.

(Lakshminarayanan et al., 2017). In the section we highlight the pitfalls of the aforementioned metrics, and demonstrate that these pitfalls can significantly affect evaluation, changing the ranking of the methods.

**Notation** We consider a classification problem with a dataset that consists of $N$ training and $n$ testing pairs $(x_i, y_i^*) \sim p(x, y)$, where $x_i$ is an object and $y_i^* \in \{1, \ldots, C\}$ is a discrete class label. A probabilistic classifier maps an object $x_i$ into a predictive distribution $\hat{p}(y \mid x_i)$. The predictive distribution $\hat{p}(y \mid x_i)$ of a deep neural network is typically defined by the softmax function $\hat{p}(y \mid x) = \text{Softmax}(z(x)/T)$, where $z(x)$ is a vector of logits and $T$ is a scalar parameter standing for the temperature of the predictive distribution. This scalar parameter is usually set to $T = 1$ or is tuned on a validation set (Guo et al., 2017). The maximum probability $\max_c \hat{p}(y = c \mid x_i)$ is called the confidence of a classifier $\hat{p}$ on an object $x_i$. $\mathbb{I}[\cdot]$ denotes the indicator function throughout the text.

### 3.1 LOG-LIKELIHOOD AND BRIER SCORE

The average test log-likelihood $\text{LL} = \frac{1}{n}\sum_{i=1}^{n} \log \hat{p}(y = y_i^* \mid x_i)$ is a popular metric for measuring the quality of in-domain uncertainty of deep learning models. It directly penalizes high probability scores assigned to incorrect labels and low probability scores assigned to the correct labels $y_i^*$.

LL is sensitive to the softmax temperature $T$. The temperature that has been implicitly learned during training can be far from optimal for the test data. However, a nearly optimal temperature can be found post-hoc by maximizing the log-likelihood on validation data. This approach is called temperature scaling or calibration (Guo et al., 2017). Despite its simplicity, the temperature scaling results in a notable improvement in the LL.

While ensembling techniques tend to have better temperature than single models, the default choice of $T = 1$ is still suboptimal. Comparing the LL with suboptimal temperatures—that is often the case in practice—can potentially produce an arbitrary ranking of different methods.

> *Comparison of the log-likelihood should only be performed at the optimal temperature.*

Empirically, we demonstrate that the overall ordering of methods and also the best ensembling method according to the LL can vary depending on temperature $T$. While this applies to most

ensembling techniques (see Figure 10), this effect is most noticeable on experiments with data augmentation on ImageNet (Figure 1).

> *We introduce a new metric called **the calibrated log-likelihood** that is the log-likelihood at the optimal temperature.*

The calibrated log-likelihood considers a model and a post-training calibration as a unified system, targeting to measure all models in the equal conditions of optimal temperature. That allows to avoid measuring calibration error that can be eliminated by a simple temperature scaling. The metric significantly affects the results of the comparison. For example, in Figure 10 the differences between Bayesian (VI, K-FAC, SWAG, dropout) and conventional non-Bayesian networks become much less pronounced, and in most cases making conventional non-Bayesian networks match the performance of Bayesian ones (VI, K-FAC, Dropout) on ResNet110, ResNet164, and WideResNet.

We show how to obtain an unbiased estimate of the calibrated log-likelihood without a held-out validation set in Section 3.5.

LL also demonstrates a high correlation with accuracy ($\rho > 0.86$), that in case of calibrated LL becomes even stronger ($\rho > 0.95$). That suggests that while (calibrated) LL measures the uncertainty of the model, it still has a significant dependence on the accuracy and vice versa. A model with higher accuracy would likely have a higher log-likelihood. See Figure 9 in Appendix C for more details.

Brier score $BS = \frac{1}{n}\frac{1}{C}\sum_{i=1}^{n}\sum_{c=1}^{C}(\mathbb{I}[y_i^* = c] - \hat{p}(y = c \,|\, x_i))^2$ has also been known for a long time as a metric for verification of predicted probabilities (Brier, 1950). Similarly to the log-likelihood, the Brier score penalizes low probabilities assigned to correct predictions and high probabilities assigned to wrong ones. It is also sensitive to the temperature of the softmax distribution and behaves similarly to the log-likelihood. While these metrics are not strictly equivalent, they show a high empirical correlation for a wide range of models on CIFAR-10, CIFAR-100 and ImageNet datasets (see Figure 8 in Appendix C ).

## 3.2 MISCLASSIFICATION DETECTION

Detection of wrong predictions of the model, or misclassifications, is a popular downstream problem relevant to the problem of in-domain uncertainty estimation. Since misclassification detection is essentially a binary classification problem, some papers measure its quality using conventional metrics for binary classification such as AUC-ROC and AUC-PR (Malinin & Gales, 2018; Cui et al., 2019; Możejko et al., 2018). These papers use an uncertainty criterion like confidence or predictive entropy $\mathcal{H}[\hat{p}(y \,|\, x_i)]$ as a prediction score.

While these metrics can be used to assess the misclassification detection performance of a single model, they cannot be used to directly compare misclassification performance across different models. Correct and incorrect predictions are specific for every model, therefore, every model induces its own binary classification problem. The induced problems can differ significantly, since different models produce different confidences and misclassify different objects. In other words, comparing such metrics implies a comparison of performance of classifiers that solve different classification problems. Such metrics are therefore incomparable.

> *AUCs for misclassification detection*
> *cannot be directly compared between different models.*

While comparing AUCs is incorrect in the setting of misclassification detection, it is correct to compare these metrics in many out-of-domain data detection problems. In that case, both objects and targets of the induced binary classification problems remain the same for all models. All out-of-domain objects have a positive label and all in-domain objects have a negative label. Note that this condition does not necessarily hold in the problem of detection of adversarial attacks. Different models generally have different inputs after an adversarial attack, so such AUC-based metrics might still be flawed.

### 3.3 CLASSIFICATION WITH REJECTION

Accuracy-confidence curves are another way to measure the performance of misclassification detection. These curves measure the accuracy on the set of objects with confidence $\max_c \hat{p}(y = c \mid x_i)$ above a certain threshold $\tau$ (Lakshminarayanan et al., 2017) and ignoring or *rejecting* the others.

The main problem with accuracy-confidence curves is that they rely too much on calibration and the actual values of confidence. Models with different temperatures have different numbers of objects at each confidence level which does not allow for a meaningful comparison. To overcome this problem, one can swhich from thresholding by the confidence level to thresholding by the number of rejected objects. The corresponding curves are then less sensitive to temperature scaling and thus allow to compare the rejection ability in a more meaningful way. Such curves have been known as *accuracy-rejection curves* (Nadeem et al., 2009). In order to obtain a scalar metric for easy comparisons, one can compute the area under this curve, resulting in AU-ARC (Nadeem et al., 2009).

### 3.4 CALIBRATION METRICS

Informally speaking, a probabilistic classifier is calibrated if any predicted class probability is equal to the true class probability according to the underlying data distribution (see (Vaicenavicius et al., 2019) for formal definitions). Any deviation from the perfect calibration is called miscalibration. For brevity, we will use $\hat{p}_{i,c}$ to denote $\hat{p}(y = c \mid x_i)$ in the current section.

Expected calibration error (ECE) (Naeini et al., 2015) is a metric that estimates model miscalibration by binning the assigned probability scores and comparing them to average accuracies inside these bins. Assuming $B_m$ denotes the $m$-th bin and $M$ is overall number of bins, the ECE is defined as follows:

$$\text{ECE} = \sum_{m=1}^{M} \frac{|B_m|}{n} \left| \text{acc}(B_m) - \text{conf}(B_m) \right|, \tag{1}$$

where $\text{acc}(B) = |B|^{-1} \sum_{i \in B} \mathbb{I}[\arg\max_c \hat{p}_{i,c} = y_i^*]$ and $\text{conf}(B) = |B|^{-1} \sum_{i \in B} \hat{p}_{i,y_i^*}$.

A recent line of works on measuring calibration in deep learning (Vaicenavicius et al., 2019; Kumar et al., 2019; Nixon et al., 2019) outlines several problems of the ECE score. Firstly, ECE is a biased estimate of the true calibration. Secondly, ECE-like scores cannot be optimized directly since they are minimized by a model with constant uniform predictions, making the infinite temperature $T = +\infty$ its global optimum. Thirdly, ECE only estimates miscalibration in terms of the maximum assigned probability whereas practical applications may require the full predicted probability vector to be calibrated. Finally, biases of ECE on different models may not be equal, rendering the miscalibration estimates incompatible. Similar concerns are also discussed by Ding et al. (2019).

Thresholded adaptive calibration error (TACE) was proposed as a step towards solving some of these problems (Nixon et al., 2019). TACE disregards all predicted probabilities that are less than a certain threshold (hence *thresholded*), chooses the bin locations adaptively so that each bin has the same number of objects (hence *adaptive*), and estimates miscalibration of probabilties across all classes in the prediction (not just the top-1 predicted class as in ECE). Assuming that $B_m^{\text{TA}}$ denotes the $m$-th thresholded adaptive bin and $M$ is the overall number of bins, TACE is defined as follows:

$$\text{TACE} = \frac{1}{CM} \sum_{c=1}^{C} \sum_{m=1}^{M} \frac{|B_m^{\text{TA}}|}{n} \left| \text{objs}(B_m^{\text{TA}}, c) - \text{conf}(B_m^{\text{TA}}, c) \right|, \tag{2}$$

where $\text{objs}(B^{\text{TA}}, c) = |B^{\text{TA}}|^{-1} \sum_{i \in B^{\text{TA}}} \mathbb{I}[y_i^* = c]$ and $\text{conf}(B^{\text{TA}}, c) = |B^{\text{TA}}|^{-1} \sum_{i \in B^{\text{TA}}} \hat{p}_{i,c}$.

Although TACE does solve several problems of ECE and is useful for measuring calibration of a specific model, it still cannot be used as a reliable criterion for comparing different models. Theory suggests that it is still a biased estimate of true calibration with different bias for each model (Vaicenavicius et al., 2019). In practice, we find that TACE is sensitive to its two parameters, the number of bins and the threshold, and does not provide a consistent ranking of different models, as shown in Figure 2.

### 3.5 CALIBRATED LOG-LIKELIHOOD AND TEST-TIME CROSS-VALIDATION

There are two common ways to perform temperature scaling using a validation set when training on datasets that only feature public training and test sets (e.g. CIFARs). The public training set might

be divided into a smaller training set and validation set, or the public test set can be split into test and validation parts (Guo et al., 2017; Nixon et al., 2019). The problem with the first method is that the resulting models cannot be directly compared with all the other models that have been trained on the full training set. The second approach, however, provides an unbiased estimate of metrics such as log-likelihood and Brier score but introduces more variance.

In order to reduce the variance of the second approach, we perform a "test-time cross-validation". We randomly divide the test set into two equal parts, then compute metrics for each half of the test set using the temperature optimized on another half. We repeat this procedure five times and average the results across different random partitions to reduce the variance of the computed metrics.

## 4 A STUDY OF ENSEMBLING & DEEP ENSEMBLE EQUIVALENT

Ensembles of deep neural networks have become a de-facto standard for uncertainty estimation and improving the quality of deep learning models (Hansen & Salamon, 1990; Krizhevsky et al., 2009; Lakshminarayanan et al., 2017). There are two main directions of training ensembles of DNNs: training stochastic computation graphs and obtaining separate snapshots of neural network parameters.

Methods based on the paradigm of **stochastic computation graphs** introduce some kind of random noise over the weights or activations of deep learning models. When the model is trained, each sample of the noise corresponds to a member of the ensemble. During test time, the predictions are averaged across the noise samples. These methods include (test-time) data augmentation, dropout (Srivastava et al., 2014; Gal & Ghahramani, 2016), variational inference (Blundell et al., 2015; Kingma et al., 2015; Louizos & Welling, 2017), batch normalization (Ioffe & Szegedy, 2015; Teye et al., 2018; Atanov et al., 2019), Laplace approximation (Ritter et al., 2018) and many more.

**Snapshot-based methods** aim to obtain sets of weights for deep learning models and then to average the predictions across these weights. The weights can be trained independently (e.g. deep ensembles (Lakshminarayanan et al., 2017)), collected on different stages of a training trajectory (e.g. snapshot ensembles (Huang et al., 2017) and fast geometric ensembles (Garipov et al., 2018)), or obtained from a sampling process (e.g. MCMC-based methods (Welling & Teh, 2011; Zhang et al., 2019)). These two paradigms can be combined. Some works suggest construction of ensembles of stochastic computation graphs (Tomczak et al., 2018), while others make use of the collected snapshots to construct a stochastic computation graph (Wang et al., 2018; Maddox et al., 2019).

In this paper we consider the following ensembling techniques: deep ensembles (Lakshminarayanan et al., 2017), snapshot ensembles (SSE by Huang et al. (2017)), fast geometric ensembling (FGE by Garipov et al. (2018)), SWA-Gaussian (SWAG by Maddox et al. (2019)), cyclical SGLD (cSGLD by Zhang et al. (2019)), variational inference (VI by Blundell et al. (2015)), K-FAC Laplace approximation (Ritter et al., 2018), dropout (Srivastava et al., 2014) and test-time data augmentation (Krizhevsky et al., 2009). These techniques were chosen to cover a diverse set of approaches keeping their predictive performance in mind.

All these techniques can be summarized as distributions $q_m(\omega)$ over parameters $\omega$ of computation graphs, where $m$ stands for the technique. During testing, one can average the predictions across parameters $\omega \sim q_m(\omega)$ to approximate the predictive distribution

$$\hat{p}(y_i \,|\, x_i) \approx \int p(y_i \,|\, x_i, \omega) q_m(\omega) \, d\omega \simeq \frac{1}{K} \sum_{k=1}^{K} p(y_i \,|\, x_i, \omega_k), \quad \omega_k \sim q_m(\omega) \tag{3}$$

For example, a deep ensemble of $S$ networks can be represented in this form as a mixture of $S$ Dirac's deltas $q_{\text{DE}}(\omega) = \frac{1}{S} \sum_{s=1}^{S} \delta(\omega - \omega_s)$, centered at independently trained snapshots $\omega_s$. Similarly, a Bayesian neural network with a fully-factorized Gaussian approximate posterior distribution over the weight matrices and convolutional kernels $\omega$ is represented as $q_{\text{VI}}(\omega) = \mathcal{N}(\omega \,|\, \mu, \text{diag}(\sigma^2))$, $\mu$ and $\sigma^2$ being the optimal variational means and variances respectively.

If one considers data augmentation as a part of the computational graph, it can be parameterized by the coordinates of the random crop and the flag for whether to flip the image horizontally or not. Sampling from the corresponding $q_{\text{aug}}(\omega)$ would generate different ways to augment the data during inference. However, as data augmentation is present by default during the training of all othe

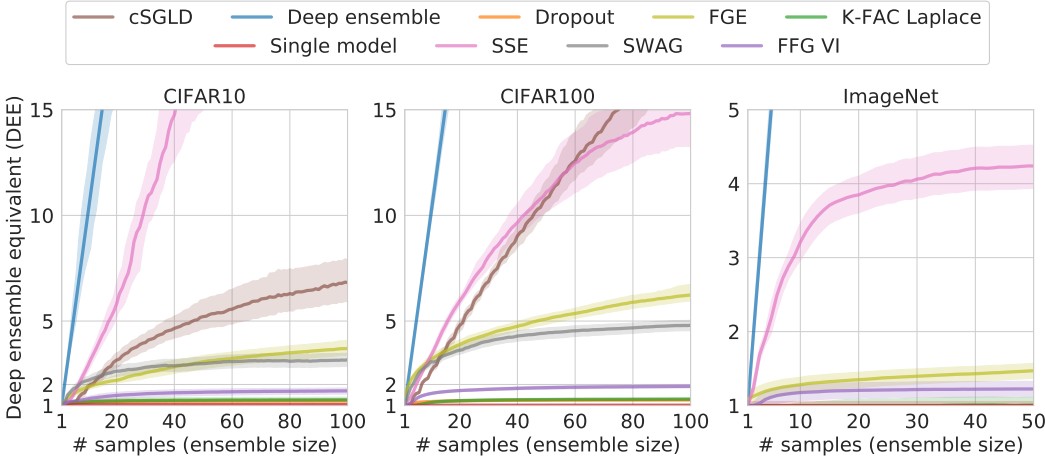

Figure 3: The deep ensemble equivalent score (DEE) for different numbers of samples on CIFAR-10, CIFAR-100, and ImageNet datasets averaged across different deep convolutional architectures. A deep ensemble equivalent score (DEE) of a model is equal to the minimum size of a deep ensemble (an ensemble of independently train networks) that achieves the same performance as the model under consideration. The score is measured in the number of models (higher is better). The area between average lower and upper bounds of DEE is shaded. **The plot demonstrates that all of the ensembling techniques are far less efficient than deep ensembles during inference and fail to produce the same level of performance as deep ensembles.** The comparison that is normalized on training time is presented in Appendix A.

mentioned ensembling techniques, it is suitable to study it in combination with these methods and not as a separate ensembling technique. We perform such an evaluation in Section 4.3.

Typically, the approximation (equation 3) requires $K$ independent forward passes through a neural network, making the test-time budget directly comparable across all methods.

## 4.1 DEEP ENSEMBLE EQUIVALENT

Most ensembling techniques under consideration are either bounded to a single mode, or provide positively correlated samples. Deep ensemble, on the other hand, is a simple technique that provides independent samples from different modes of the loss landscape, which, intuitively, should result in a better ensemble. Therefore deep ensembles can be considered a strong baseline for the performance of other ensembling techniques given a fixed test-time computation budget.

Comparing the performance of ensembling techniques is, however, a challenging problem. Different models on different datasets achieve different values of metrics; their dependence on the number of samples is non-trivial, and varies depending on a specific model and dataset. Values of the metrics are thus lacking in interpretability as the gain in performance has to be compared against a model- and dataset-specific baseline.

Aiming to introduce perspective and interpretability in our study, we introduce the *deep ensemble equivalent* score that employs deep ensembles to measure the performance of other ensembling techniques. Specifically, the deep ensemble equivalent score answers the following question:

*What size of deep ensemble yields the same performance as a particular ensembling method?*

Following the insights from the previous sections, we base the deep ensemble equivalent on the calibrated log-likelihood (CLL). Formally speaking, we define the deep ensemble equivalent (DEE) for an ensembling method $m$ and its upper and lower bounds as follows:

$$\mathrm{DEE}_m(k) = \min\left\{l \in \mathbb{R}, l \geq 1 \,\middle|\, \mathrm{CLL}_{DE}^{\mathrm{mean}}(l) \geq \mathrm{CLL}_m^{\mathrm{mean}}(k)\right\}, \quad (4)$$

$$\mathrm{DEE}_m^{\substack{\mathrm{upper} \\ \mathrm{lower}}}(k) = \min\left\{l \in \mathbb{R}, l \geq 1 \,\middle|\, \mathrm{CLL}_{DE}^{\mathrm{mean}}(l) \mp \mathrm{CLL}_{DE}^{\mathrm{std}}(l) \geq \mathrm{CLL}_m^{\mathrm{mean}}(k)\right\}, \quad (5)$$

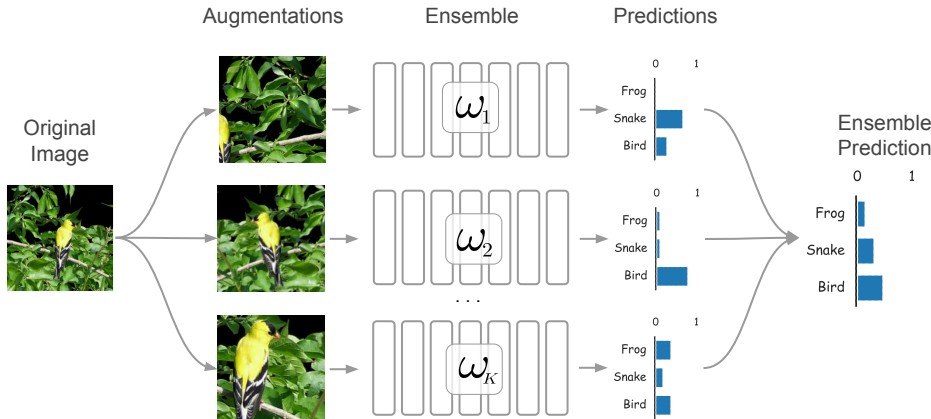

Figure 4: An illustration of test-time augmentation (TTA) for an ensemble. We apply every member of an ensemble to a separate random augmentation of an image. The predictions of all members are averaged to produce a final prediction of an ensemble. In our experiments, TTA leads to a significant boost of the performance for most of the ensembling techniques on ImageNet with a sufficient computational budget (see Figure 5).

where $\text{CLL}_m^{\text{mean/std}}(l)$ are the mean and the standard deviation of the calibrated log-likelihood achieved by an ensembling method $m$ with $l$ samples. We compute $\text{CLL}_{\text{DE}}^{\text{mean}}(l)$ and $\text{CLL}_{\text{DE}}^{\text{std}}(l)$ for natural numbers $l \in \mathbb{N}_{>0}$ and use linear interpolation to define them for real values $l \geq 1$. In the following plots we report $\text{DEE}_m(k)$ for different methods $m$ with different numbers of samples $k$, and shade the area between the respective lower and upper bounds $\text{DEE}_m^{\text{lower}}(k)$ and $\text{DEE}_m^{\text{upper}}(k)$.

## 4.2 EXPERIMENTS

We compute the deep ensemble equivalent (DEE) of various ensembling techniques for four popular deep architectures: VGG16 (Simonyan & Zisserman, 2014), PreResNet110/164 (He et al., 2016), and WideResNet28x10 (Zagoruyko & Komodakis, 2016) on CIFAR-10/100 datasets (Krizhevsky et al., 2009), and ResNet50 (He et al., 2016) on ImageNet dataset (Russakovsky et al., 2015). We use PyTorch (Paszke et al., 2017) for implementation of these models, building upon available public implementations. Our implementation closely matches the quality of methods that has been reported in original works. Technical details on training, hyperparameters and implementations can be found in Appendix B. The source code and all computed metrics are available on GitHub[1].

As one can see on Figure 3, ensembling methods clearly fall into three categories. SSE and cSGLD outperform all other techniques except deep ensembles and enjoy a near-linear scaling of DEE with the number of samples on CIFAR datasets. The investigation of weight-space trajectories of cSGLD and SSE (Huang et al., 2017; Zhang et al., 2019) suggests that these methods can efficiently explore different modes of the loss landscape. In terms of the deep ensemble equivalent, these methods do not saturate unlike other methods that are bound to a single mode. We found SSE to still saturate on ImageNet. This is likely due to suboptimal hyperparameters of the cyclic learning rate schedule. More verbose results are presented in Figures 11–13 and in Table 5 and Table 8 in Appendix C.

In our experiments SSE typically outperforms cSGLD. This is mostly due to the fact that SSE has a much larger training budget. The cycle lengths and learning rates of SSE and cSGLD are comparable, however, SSE collects one snapshot per cycle while cSGLD collects three snapshots. This makes samples from SSE less correlated with each other while increasing the training budget threefold. Both SSE and cSGLD can be adjusted to obtain a different trade-off between the training budget and the DEE-to-samples ratio. We reused the schedules provided in the original papers (Huang et al., 2017; Zhang et al., 2019).

---

[1] Source code: `https://github.com/bayesgroup/pytorch-ensembles`

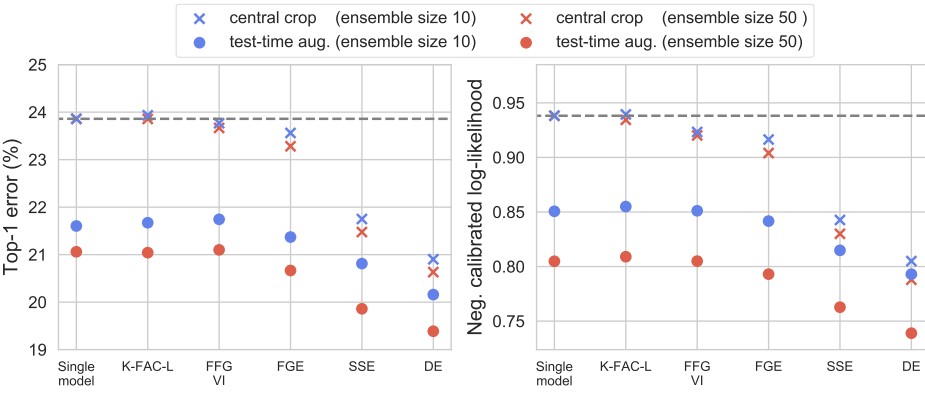

Figure 5: How to read results: $\times \xrightarrow[\text{10 samples}]{\text{test-time aug}} \bullet$, $\times \xrightarrow[\text{50 samples}]{\text{test-time aug}} \bullet$. The negative calibrated log-likelihood (lower is better) for different ensembling techniques on ImageNet. We report performance for two regimes. *Central-crop evaluation* ($\times\times$) means every member of an ensemble is applied to a central crop of an image, and *test-time data augmentation* ($\bullet\bullet$) means each member of the ensemble is applied to a separate random augmentation of the image. **Test-time data augmentation significantly improves ensembles with no additional computational cost.** Interestingly, a single model with TTA performs competitively with methods that require significantly larger parameters budget, and training complexity, e.g., a single model with TTA performs closely to pure deep ensembles (DE) of the same size.

Being more "local" methods, FGE and SWAG perform worse than SSE and cSGLD, but still significantly outperform "single-snapshot" methods like dropout, K-FAC Laplace approximation and variational inference. We hypothesize that by covering a single mode with a set of snapshots, FGE and SWAG provide a better fit for the local geometry than models trained as stochastic computation graphs. This implies that the performance of FGE and SWAG should be achievable by single-snapshot methods. However, one might need more elaborate posterior approximations and better inference techniques in order to match the performance of FGE and SWAG by training a stochastic computation graph end-to-end (as opposed to SWAG that constructs a stochastic computation graph post-hoc).

The deep ensemble equivalent curves allow us to notice the common behaviour of different methods, e.g. the relation between deep ensembles, snapshot methods, advanced local methods and single-snapshot local methods. They also allow us to notice inconsistencies that may indicate a suboptimal choice of hyperparameters. For example, we find that SSE on ImageNet quickly saturates, unlike SSE on CIFAR datasets (Figure 3). This may indicate that the hyperparameters used on ImageNet are not good enough for efficient coverage of different modes of the loss landscape. We also find that SSE on WideResNet on CIFAR-10 achieves a DEE score of 100 on approx. 70 samples (Figure 12). This may indicate that the members of the deep ensemble for this dataset-architecture pair are underfitted and may benefit from longer training or a different learning rate schedule. Such inconsistencies might be more difficult to spot using plain calibrated log-likelihood plots.

### 4.3 TEST-TIME DATA AUGMENTATION IMPROVES ENSEMBLES FOR FREE

Data augmentation is a time-honored technique that is widely used in deep learning, and is a crucial component for training modern DNNs. Test-time data augmentation has been used for a long time to improve the performance of convolutional networks. For example, multi-crop evaluation has been a standard procedure for the ImageNet challenge (Simonyan & Zisserman, 2014; Szegedy et al., 2015; He et al., 2016). It is, however, often overlooked in the literature on ensembling techniques in deep learning. In this section, we study the effect of test-time data augmentation on the aforementioned ensembling techniques. To keep the test-time computation budget the same, we sample one random augmentation for each member of an ensemble. Figure 5 reports the calibrated log-likelihood on combination of ensembles and test-time data augmentation for ImageNet. Other metrics and results on CIFAR-10/100 datasets are reported in Appendix C. We have used the standard data augmen-

tation: random horizontal flips and random padded crops for CIFAR-10/100 datasets, and random horizontal flips and random resized crops for ImageNet (see more details in Appendix B).

Test-time data augmentation (Figure 4) consistently improves most ensembling methods, especially on ImageNet, where we see a clear improvement across all methods (Figure 5 and Table 7). The performance gain for powerful ensembles (deep ensembles, SSE and cSGLD) on CIFAR datasets is not as dramatic (Figures 14–15 and Table 4). This is likely due to the fact that CIFAR images are small, making data augmentation limited, whereas images from ImageNet allow for a large number of diverse samples of augmented images. On the other hand, while the performance of "single-snapshot" methods (e.g. variational inference, K-FAC Laplace and dropout) is improved significantly, they perform approximately as good as an augmented version of a *single* model across all datasets.w

Interestingly, test-time data augmentation on ImageNet improves accuracy but decreases the uncalibrated log-likelihood of deep ensembles (Table 7 in Appendix C). Test-time data augmentation breaks the nearly optimal temperature of deep ensembles and requires temperature scaling to reveal the actual performance of the method, as discussed in Section 3.1. The experiment demonstrates that ensembles may be highly miscalibrated by default while still providing superior predictive performance after calibration.

We would like to note that test-time data augmentation does not always break the calibration of an ensemble, and, on the contrary, test-time data augmentation often improves the calibration of an ensemble. In our experiments, decalibration was caused by the extreme magnitude of a random crop, that is conventionally used for ImageNet augmentation. **Using less extreme magnitude of the random crop fixes decalibration**, that makes test-time data augmentation a more practical method that provides out-of-the-box calibration. Although, as we demonstrated earlier, there is no guarantee that any ensemble is calibrated out-of-the-box. If we are willing to apply post-hoc calibration, the final performance can be much better with more severe augmentations.

## 5    DISCUSSION & CONCLUSION

We have explored the field of in-domain uncertainty estimation and performed an extensive evaluation of modern ensembling techniques. Our main findings can be summarized as follows:

- Temperature scaling is a must even for ensembles. While ensembles generally have better calibration out-of-the-box, they are not calibrated perfectly and can benefit from the procedure. A comparison of log-likelihoods of different ensembling methods without temperature scaling might not provide a fair ranking, especially if some models happen to be miscalibrated.

- Many common metrics for measuring in-domain uncertainty are either unreliable (ECE and analogues) or cannot be used to compare different methods (AUC-ROC, AUC-PR for misclassification detection; accuracy-confidence curves). In order to perform a fair comparison of different methods, one needs to be cautious of these pitfalls.

- Many popular ensembling techniques require dozens of samples for test-time averaging, yet are essentially equivalent to a handful of independently trained models. Deep ensembles dominate other methods given a fixed test-time budget. The results indicate, in particular, that exploration of different modes in the loss landscape is crucial for good predictive performance.

- Methods that are stuck in a single mode are unable to compete with methods that are designed to explore different modes of the loss landscape. Would more elaborate posterior approximations and better inference techniques shorten this gap?

- Test-time data augmentation is a surprisingly strong baseline for in-domain uncertainty estimation. It can significantly improve other methods without increasing training time or model size since data augmentation is usually already present during training.

Our takeaways are aligned with the take-home messages of Ovadia et al. (2019) that relate to in-domain uncertainty estimation. We also observe a stable ordering of different methods in our experiments, and observe that deep ensembles with few members outperform methods based on stochastic computation graphs.

A large number of unreliable metrics inhibits a fair comparison of different methods. Because of this, we urge the community to aim for more reliable benchmarks in the numerous setups of uncertainty estimation.

ACKNOWLEDGMENTS

Dmitry Vetrov and Dmitry Molchanov were supported by the Russian Science Foundation grant no. 19-71-30020. This research was supported in part through computational resources of HPC facilities at NRU HSE.

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

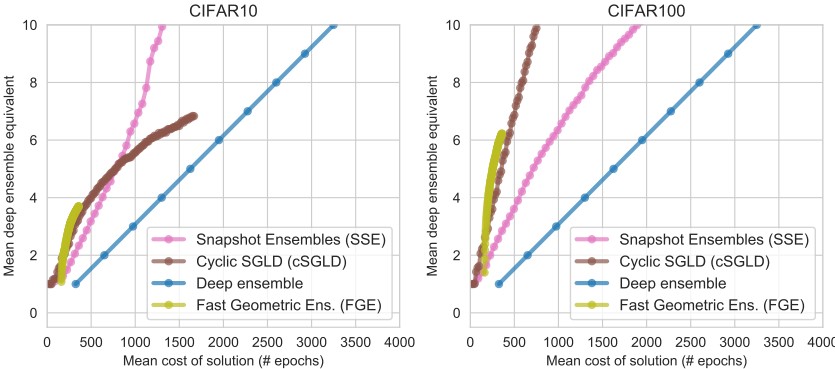

Figure 6: The mean cost of training of an ensemble vs. the mean deep ensemble equivalent score. Each marker on the plot denotes one snapshot of weights.

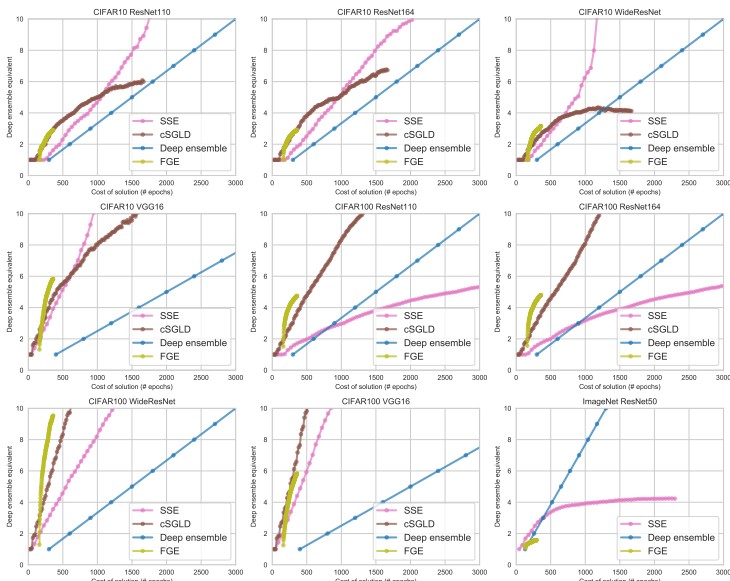

Figure 7: Cost of training of an ensemble vs. its quality (DEE). Each marker on a plot denotes one snapshot of weights.

## A    IS "EFFICIENT" TRAINING OF ENSEMBLES EFFICIENT AT ALL?

**Yes! If you care most about training time efficiency.** All snapshot based methods (SSE, cSGLD and FGE) on average (Figure 6) tend to achieve the same performance as deep ensembles using 2-5× less training epochs on CIFAR datasets.

**The gain comes at a cost of inference efficiency and memory consumption.** "Efficient" snapshot-based methods require to store much more samples of weights compared to deep ensembles making inference significantly more expensive (up to ×25) given the same predictive performance.

**You need to get lucky with hyperparameter choice.** While on average "efficient" snapshot-based methods require less training resources, they might be completely inefficient if your hyperparameter choice is sub-optimal (see Figure 7). Such hyperparameters as the maximum learning rate, the length of learning rate decay cycle, the snapshot saving schedule can all impact the performance significantly.

This analysis assumes that we use only conventional training procedure. Many models can most likely be trained, stored and executed much more efficiently with methods like super-convergence, stochastic weight averaging, compression, distillation and others. These methods are out of the scope of the paper but are interesting topics for future research. The current choice of hyperparameters may also not be optimal. We reuse the hyperparameters used in the original papers.

# B EXPERIMENTAL DETAILS

Implementations of deep ensembles, SWAG, FGE and K-FAC Laplace are heavily based on the original PyTorch implementations of stochastic weight averaging (SWA) [2] and SWAG [3]. Implementations of cyclical MCMC and snapshot ensembles are based on the original implementation of cyclical MCMC [4]. We hypothesize that the optimal hyperparameters of ensembling methods may vary widely depending on the computational budget and the number of samples in the ensemble. Searching for the optimal values for each configuration is outside the scope of this paper so we stick to the originally proposed hyperparameters whenever possible.

**Implied probabilistic model** Conventional neural networks for classification are usually trained using the average cross-entropy loss function with weight decay regularization hidden inside an optimizer in a deep learning framework like PyTorch. The underlying optimization problem can be written as follows:

$$L(w) = -\frac{1}{N} \sum_{i=1}^{N} \log \hat{p}(y_i^* \,|\, x_i, w) + \frac{\lambda}{2} \|w\|^2 \to \min_w, \qquad (6)$$

where $\{(x_i, y_i^*)\}_{i=1}^N$ is the training dataset of $N$ objects $x_i$ with corresponding labels $y_i^*$, $\lambda$ is the weight decay scale and $\hat{p}(j \,|\, x_i, w)$ denotes the probability that a neural network with parameters $w$ assigns to class $j$ when evaluated on object $x_i$.

The cross-entropy loss defines a likelihood function $p(y^* \,|\, x, w)$ and weight decay regularization, or $L_2$ regularization corresponding to a certain Gaussian prior distribution $p(w)$. The whole optimization objective then corresponds to the *maximum a posteriori* inference in the following probabilistic model:

$$p(y^*, w \,|\, x) = p(y^* \,|\, x, w) p(w), \qquad (7)$$

$$\log p(y^* \,|\, x, w) = \log \prod_{i=1}^{N} p(y_i^* \,|\, x_i, w) = \sum_{i=1}^{N} \log \hat{p}(y_i^* \,|\, x_i, w), \qquad (8)$$

$$\log p(w) = \frac{-N\lambda}{2} \|w\|^2 + \text{const} \iff p(w) = \mathcal{N}\left(w \,\big|\, 0, (N\lambda)^{-1}I\right) \qquad (9)$$

In order to make the results comparable across all ensembling techniques, we used the same probabilistic model for all methods, choosing fixed weight decay parameters for each architecture. We used the softmax-based likelihood for all models. We also use the fully-factorized zero-mean Gaussian prior distribution with variances $\sigma^2 = (N\lambda)^{-1}$ where the number of objects $N$ and the weight decay scale $\lambda$ are dictated by the particular datasets and neural architectures as defined in the following paragraph.

**Conventional networks** To train a single network on CIFAR-10/100, we used SGD with batch size of 128, momentum 0.9 and model-specific parameters, i.e. the initial learning rate ($lr_{init}$), the weight decay coefficient (wd), and the number of optimization epochs (epoch). Specific hyperparameters are shown in Table 1. The models were trained with a unified learning rate scheduler that is shown in equation 10. All models have been trained using data augmentation that consists of horizontal flips and a random crop of 32 pixels with a padding of 4 pixels[5]. The standard data normalization has also been applied. Weight decays, initial learning rates, and the learning rate scheduler were taken from (Garipov et al., 2018) paper. Compared with hyperparameters of (Garipov et al., 2018), we increased the number of optimization epochs since we found that all models were underfitted. While the original WideResNet28x10 network includes a number of dropout layers with $p = 0.3$ and is trained for 200 epochs, we find that the WideResNet28x10 underfits in this setting and requires a longer training. Therefore, we used $p = 0$ which reduces training time while bearing

---

[2]`https://github.com/timgaripov/swa`
[3]`https://github.com/wjmaddox/swa_gaussian`
[4]`https://github.com/ruqizhang/csgmcmc/tree/master/experiments`
[5]`Compose([RandomHorizontalFlip(), RandomCrop(32, padding=4)])`

| Model | $lr_{init}$ | epochs | wd |
|---|---|---|---|
| VGG | 0.05 | 400 | 5e-4 |
| PreResNet110 | 0.1 | 300 | 3e-4 |
| PreResNet164 | 0.1 | 300 | 3e-4 |
| WideResNet28x10 | 0.1 | 300 | 5e-4 |

Table 1: Hyperparameters of models trained on CIFARs for single-model evaluation

no significant effect on final model performance in our experiments.

$$\mathtt{lr}(i) = \begin{cases} lr_{init}, & i \in [0, 0.5 \cdot \text{epochs}] \\ lr_{init} \cdot (1.0 - 0.99 * (i/\text{epochs} - 0.5)/0.4), & i \in [0.5 \cdot \text{epochs}, 0.9 \cdot \text{epochs}] \\ lr_{init} \cdot 0.01, & \text{otherwise} \end{cases} \quad (10)$$

On ImageNet dataset we used ResNet50 with default hyperparameters taken from PyTorch examples [6]. Specifically, we used SGD with momentum 0.9, batch size of 256, initial learning rate 0.1, weight decay $1e - 4$. Training included data augmentation[7] (scaling, random crops of size $224 \times 224$, horizontal flips), normalization and learning rate scheduler $lr = lr_{init} \cdot 0.1^{\text{epoch}//30}$ where $//$ denotes integer division. We only deviated from standard parameters by increasing the number of training epochs from 90 to 130. Or models achieve top-1 error of $23.81 \pm 0.15$ that closely matches the accuracy of ResNet50 provided by PyTorch which is 23.85 [8]. Training of one model on a single NVIDIA Tesla V100 GPU takes approximately 5.5 days.

**Deep ensembles** Deep ensembles (Lakshminarayanan et al., 2017) average the predictions across networks trained independently starting from different initializations. To obtain a deep ensemble we repeat the described procedure of training standard networks 128 times for all architectures on CIFAR-10 and CIFAR-100 datasets (1024 networks over all) and 50 times for ImageNet dataset. Every member of deep ensembles was trained with exactly the same hyperparameters as conventional models of the same architecture.

**Dropout** Binary dropout (or MC dropout) (Srivastava et al., 2014; Gal & Ghahramani, 2016) is one of the most widely known ensembling techniques. It involves putting a multiplicative Bernoulli noise with a parameter $p$ over the activations of either a fully connected layer or a convolutional layer, averaging predictions of the network w.r.t. noise at test-time. Dropout layers were applied to VGG and WideResNet networks on CIFAR-10 and CIFAR-100 datasets. Dropout for VGG was applied to fully connected layers with $p = 0.5$. Two dropout layers were applied: one before the first fully connected layer and one before the second one. While the original version of VGG for CIFARs (Zagoruyko, 2015) exploits more dropout layers, we observed that any additional dropout layer deteriorates the performance of the model in ether deterministic or stochastic mode. Dropout for WideResNet was applied in accordance with the original paper (Zagoruyko & Komodakis, 2016) with $p = 0.3$. Dropout usually increases the time needed to achieve convergence. Because of this, WideResNet networks with dropout were trained for 400 epochs instead of 300 epochs for deterministic case, and VGG networks have always been trained with dropout. All the other hyperparameters were the same as in the case of conventional models.

**Variational Inference** Variational Inference (VI) approximates the true posterior distribution over weights $p(w \mid Data)$ with a tractable variational approximation $q_\theta(w)$ by maximizing a so-called variational lower bound $\mathcal{L}$ (eq. 11) w.r.t. the parameters of variational approximation $\theta$. We used a fully-factorized Gaussian approximation $q(w)$ and Gaussian prior distribution $p(w)$.

$$\mathcal{L}(\theta) = \mathbb{E}_q \log p(y^* \mid x, w) - KL(q_\theta(w) \mid\mid p(w)) \to \max_\theta \quad (11)$$

$$q(w) = \mathcal{N}(w \mid \mu, \text{diag}(\sigma^2)) \quad p(w) = N(w \mid 0, \text{diag}(\sigma_p^2)), \quad \text{where } \sigma_p^2 = (N \cdot \text{wd})^{-1} \quad (12)$$

---

[6] `https://github.com/pytorch/examples/tree/ee964a2/imagenet`
[7] `Compose([RandomResizedCrop(224), RandomHorizontalFlip()])`
[8] `https://pytorch.org/docs/stable/torchvision/models.html`

| Architecture | Optimal noise scale | | | |
| | CIFAR-10 | CIFAR-10-aug | CIFAR-100 | CIFAR-100-aug |
| --- | --- | --- | --- | --- |
| VGG16BN | 0.042 | 0.042 | 0.100 | 0.100 |
| PreResNet110 | 0.213 | 0.141 | 0.478 | 0.401 |
| PreResNet164 | 0.120 | 0.105 | 0.285 | 0.225 |
| WideResNet28x10 | 0.022 | 0.018 | 0.022 | 0.004 |

Table 2: Optimal noise scale for K-FAC Laplace for different datasets and architectures. For ResNet50 on ImageNet, the optimal scale found was 2.0 with test-time augmentation and 6.8 without test-time augmentation.

In the case of such a prior, the probabilistic model remains consistent with the conventional training which corresponds to MAP inference in the same probabilistic model. We used variational inference for both convolutional and fully-connected layers where variances of the weights were parameterized by $\log \sigma$. For fully-connected layers we applied the local reparameterization trick (LRT, (Kingma et al., 2015)).

While variational inference provides a theoretically grounded way to approximate the true posterior, it tends to underfit deep learning models in practice (Kingma et al., 2015). The following tricks are applied to deal with it: pre-training (Molchanov et al., 2017) or equivalently annealing of $\beta$ (Sønderby et al., 2016), and scaling $\beta$ down (Kingma et al., 2015; Ullrich et al., 2017).

During pre-training we initialize $\mu$ with a snapshot of weights of a pre-trained conventional model, and initialize $\log \sigma$ with a model-specific constant $\log \sigma_{init}$. The KL-divergence – except for the term corresponding to the weight decay – is scaled with a model-specific parameter $\beta$. The weight decay term is implemented as a part of the optimizer. We used a fact that the KL-divergence between two Gaussian distributions can be rewritten as two terms, one of which is equivalent to the weight decay regularization.

On CIFAR-10 and CIFAR-100 we used $\beta$ equal to 1e-4 for VGG, ResNet100 and ResNet164 networks, and $\beta$ equal to 1e-5 for WideResNet. Log-variance $\log \sigma_{init}$ was initialized with $-5$ for all models. Parameters $\mu$ were optimized with SGD in the same manner as in the case of conventional networks except that the initial learning rate $lr_{init}$ was set to 1e-3. We used a separate Adam optimizer with a constant learning rate of 1e-3 to optimize log-variances of the weights $\log \sigma$. Pre-training was done for 300 epochs, and after that the remaining part of training was done for 100 epochs. On ImageNet we used $\beta = $ 1e-3, $lr_{init} = 0.01$, $\log \sigma_{init} = -6$, and trained the model for 45 epochs after pre-training.

**K-FAC Laplace** The Laplace approximation uses the curvature information of the appropriately scaled loss function to construct a Gaussian approximation to the posterior distribution. Ideally, one would use the Hessian of the loss function as a covariance matrix and use the maximum a posteriori estimate $w^{MAP}$ as a mean of the Gaussian approximation:

$$\log p(w \,|\, x, y^*) = \log p(y^* \,|\, x, w) + \log p(w) + \text{const} \tag{13}$$

$$w^{MAP} = \arg\max_{w} \log p(w \,|\, x, y^*); \quad \Sigma = -\nabla\nabla \log p(w \,|\, x, y^*) \tag{14}$$

$$p(w \,|\, x, y^*) \approx \mathcal{N}(w \,|\, w^{MAP}, \Sigma) \tag{15}$$

In order to keep the method scalable, we use the Fisher Information Matrix as an approximation to the true Hessian (Martens & Grosse, 2015). For K-FAC Laplace, we use the whole dataset to construct an approximation to the empirical Fisher Information Matrix, and use the $\pi$ correction to reduce the bias (Ritter et al., 2018; Martens & Grosse, 2015). Following (Ritter et al., 2018), we find the optimal noise scale for K-FAC Laplace on a held-out validation set by averaging across five random initializations. We then reuse this scale for networks trained without a hold-out validation set. We report the optimal values of scales in Table 2. Note that the optimal scale is different depending on whether we use test-time data augmentation or not. Since the data augmentation also introduces some amount of additional noise, the optimal noise scale for K-FAC Laplace with data augmentation is lower.

**Snapshot ensembles**  Snapshot ensembles (SSE) (Huang et al., 2017) is a simple example of an array of methods which collect samples from a training trajectory of a network in weight space to construct an ensemble. Samples are collected in a cyclical manner: during each cycle the learning rate goes from a large value to near-zero and snapshot of weights of the network is taken at the end of the cycle. SSE uses SGD with a cosine learning schedule defined as follows:

$$\alpha(t) = \frac{\alpha_0}{2} \left( \cos \left( \frac{\pi \bmod (t-1, \lceil T/M \rceil)}{\lceil T/M \rceil} \right) + 1 \right), \tag{16}$$

where $\alpha_0$ is the initial learning rate, $T$ is the total number of training iterations and M is the number of cycles.

For all datasets and models hyperparameters from the original SSE paper are reused. For CIFAR-10/100 length of the cycle is 40 epochs, maximum learning rate is 0.2, batch size is 64. On ResNet50 and ImageNet length of the cycle is 45 epochs, maximum learning rate is 0.1, batch size is 256.

**Cyclical SGLD**  Cyclical Stochastic Gradient Langevin Dynamics (cSGLD) (Zhang et al., 2019) is a state-of-the-art ensembling method for deep neural networks pertaining to stochastic Markov Chain Monte Carlo family of methods. It bears similarity to SSE, e.g. it employs SGD with a learning rate schedule described with the equation 16 and training is cyclical in the same manner. Its main differences from SSE are the introduction of gradient noise and the capturing of several snapshots per cycle, both of which can aid in sampling from posterior distribution over neural network weights efficiently.

Some parameters from the original paper are reused: length of cycle is 50 epochs, maximum learning rate is 0.5, batch size is 64. Number of epochs with gradient noise per cycle is 3 epochs. This was found to yield much higher predictive performance and better uncertainty estimation compared to the original paper choice of 10 epochs for CIFAR-10 and 3 epochs for CIFAR-100.

Finally, the results of cyclical Stochastic Gradient Hamiltonian Monte Carlo (SGHMC), (Zhang et al., 2019) which reportedly has marginally better performance compared with cyclical SGLD, could not be reproduced with any value of SGD momentum term. Because of this, we only include cyclical SGLD in our benchmark.

**FGE**  Fast Geometric Ensembling (FGE) is an ensembling method that is similar to SSE in that it collects weight samples from a training trajectory to construct an ensemble. Its main differences from SSE are pretraining, short cycle length and a piecewise-linear learning rate schedule:

$$\alpha(i) = \begin{cases} (1 - 2t(i))\alpha_1 + 2t(i)\alpha_2 & 0 < t(i) \leq \frac{1}{2} \\ (2 - 2t(i))\alpha_2 + (2t(i) - 1)\alpha_1 & \frac{1}{2} < t(i) \leq 1 \end{cases}. \tag{17}$$

Hyperparameters of the original implementation of FGE are reused. Model pretraining is done with SGD for 160 epochs according to the standard learning rate schedule described in equation 10 with maximum learning rates from Table 1. After that, a desired number of FGE cycles is done with one snapshot per cycle collected. For VGG the learning rate is changed with parameters $\alpha_1 = 1e-2, \alpha_2 = 5e-4$ in a cycle with cycle length of 2 epochs. For other networks the learning rate is changed with parameters $\alpha_1 = 5e-2, \alpha_2 = 5e-4$ with cycle length of 4 epochs. Batch size is 128.

**SWAG**  SWA-Gaussian (SWAG) (Maddox et al., 2019) is an ensembling method based on fitting a Gaussian distribution to model weights on the SGD training trajectory and sampling from this distribution to construct an ensemble.

Like FGE, SWAG has a pretraining stage which is done according to the standard learning rate schedule described in equation 10 with maximum learning rates from Table 1. After that, training continues with a constant learning rate of 1e-2 for all models except for PreResNet110 and PreResNet164 on CIFAR-100 where it continues with a constant learning rate of 5e-2 in accordance with the original paper. Rank of the empirical covariance matrix which is used for estimation of Gaussian distribution parameters is set to be 20.

## C  ADDITIONAL EXPERIMENTAL RESULTS

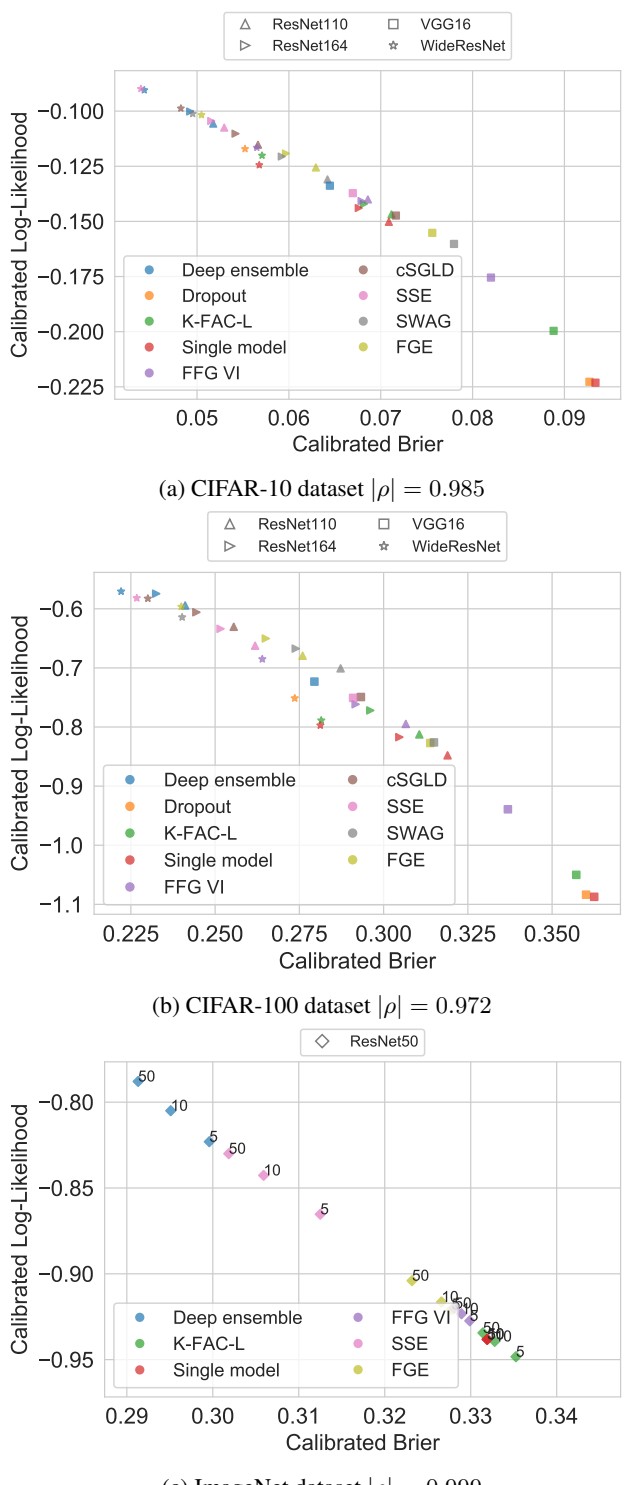

(a) CIFAR-10 dataset $|\rho| = 0.985$

(b) CIFAR-100 dataset $|\rho| = 0.972$

(c) ImageNet dataset $|\rho| = 0.999$

Figure 8: The average log-likelihood vs the Brier score on a test dataset for different ensemble methods on CIFAR-10 (a) and CIFAR-10 (b) and ImageNet (c) datasets. While not being equivalent, these metrics demonstrate a strong linear correlation. The correlation coefficient is denoted as $\rho$.

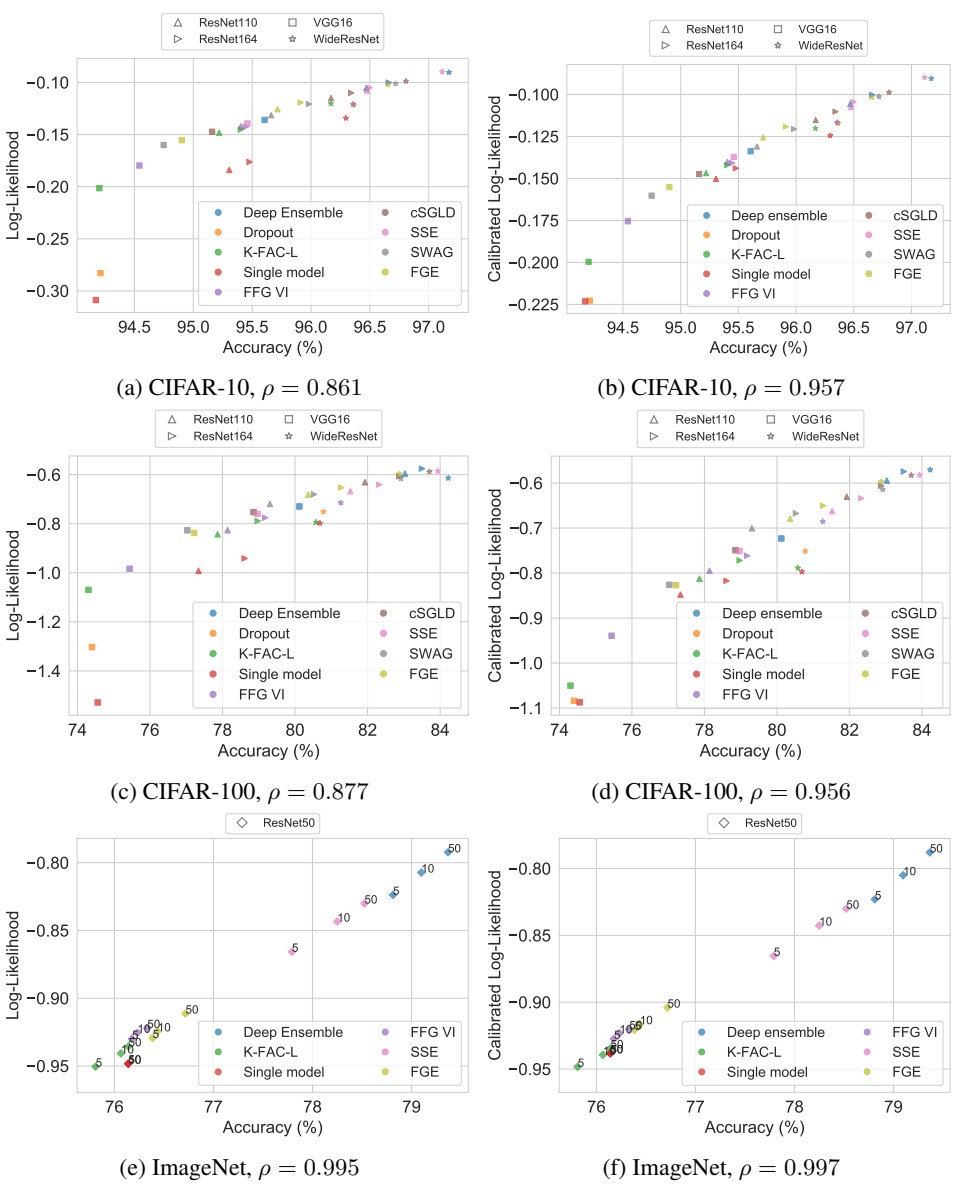

Figure 9: Log-likelihood vs accuracy for different ensembles before (a, c, e) and after (b, d, f) calibration. Both plain log-likelihood and especially calibrated log-likelihood are highly correlated with accuracy.

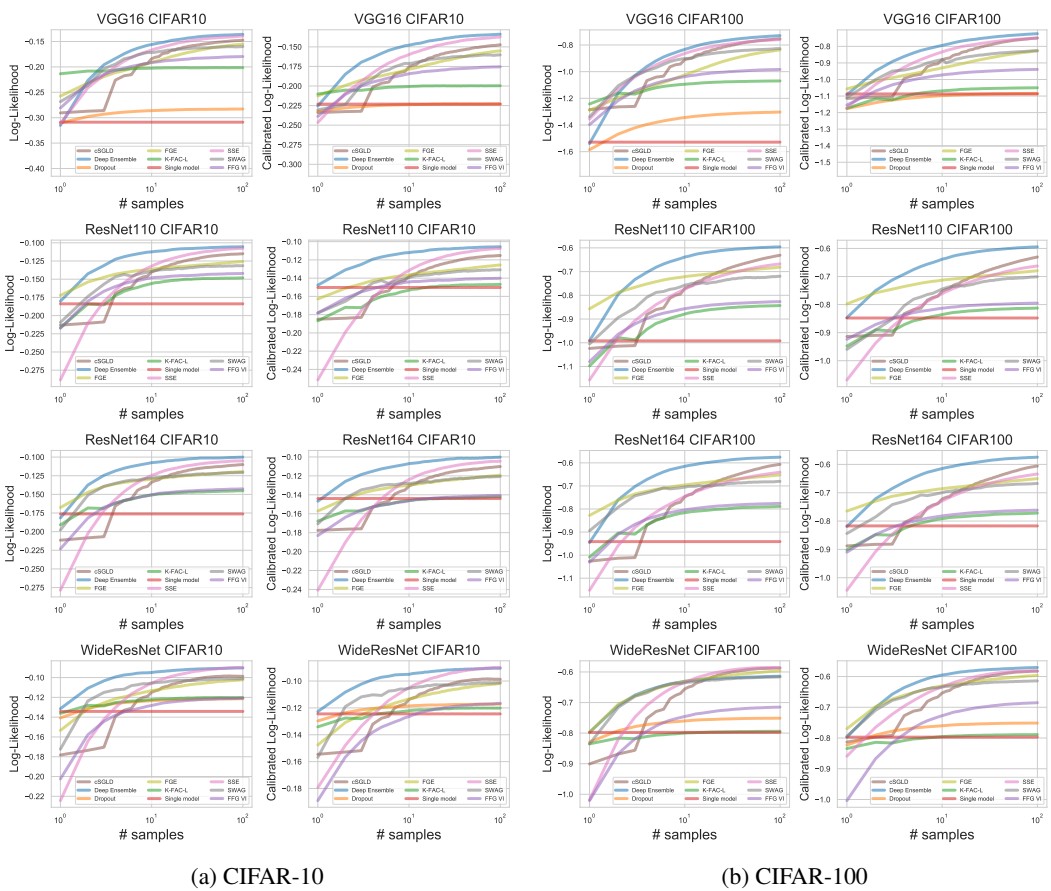

(a) CIFAR-10                                    (b) CIFAR-100

Figure 10: A side-by-side comparison of log-likelihood and calibrated log-likelihood on CIFAR-10 (a) and CIFAR-100 (b) datasets. On CIFAR-10 the performance of one network becomes close to dropout, variational inference (vi), and K-FAC Laplace approximation (kfacl) after calibration on all models except VGG. On CIFAR-100 deep ensembles move to the first position in the ranking after calibration on WideResNet and VGG. See Section 3.1 for details on the calibrated log-likelihood.

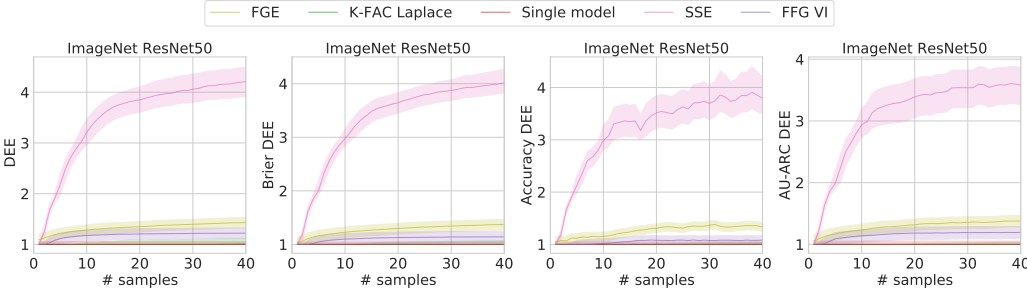

Figure 11: The deep ensemble equivalent of various ensembling techniques on ImageNet. Solid lines: mean DEE for different methods and architectures. Area between DEE[lower] and DEE[upper] is shaded. Columns 2–4 correspond to DEE based on other metrics, defined similarly to the log-likelihood-based DEE. The results are consistent for all metrics

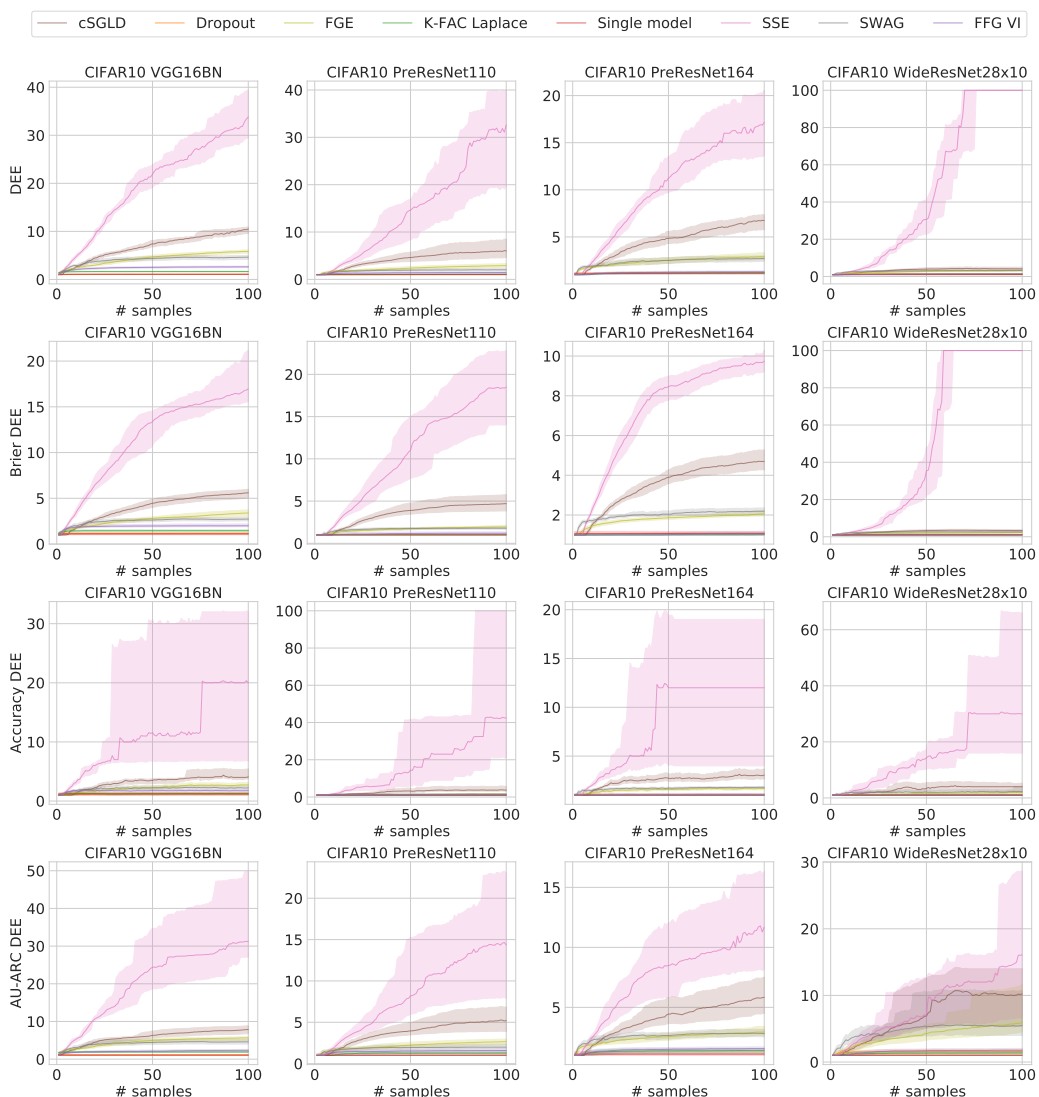

Figure 12: The deep ensemble equivalent of various ensembling techniques on CIFAR-10. Solid lines: mean DEE for different methods and architectures. Area between DEE$^{\text{lower}}$ and DEE$^{\text{upper}}$ is shaded. Lines 2–4 correspond to DEE based on other metrics, defined similarly to the log-likelihood-based DEE. Note that while the actual scale of DEE varies from metric to metric, the ordering of different methods and the overall behaviour of the lines remain the same.

SSE outperforms deep ensembles on CIFAR-10 on the WideResNet architecture. It possibly indicates that the cosine learning rate schedule and longer training of SSE are more suitable for this architecture than the piecewise-linear learning rate schedule and the number of epochs used in deep ensembles.

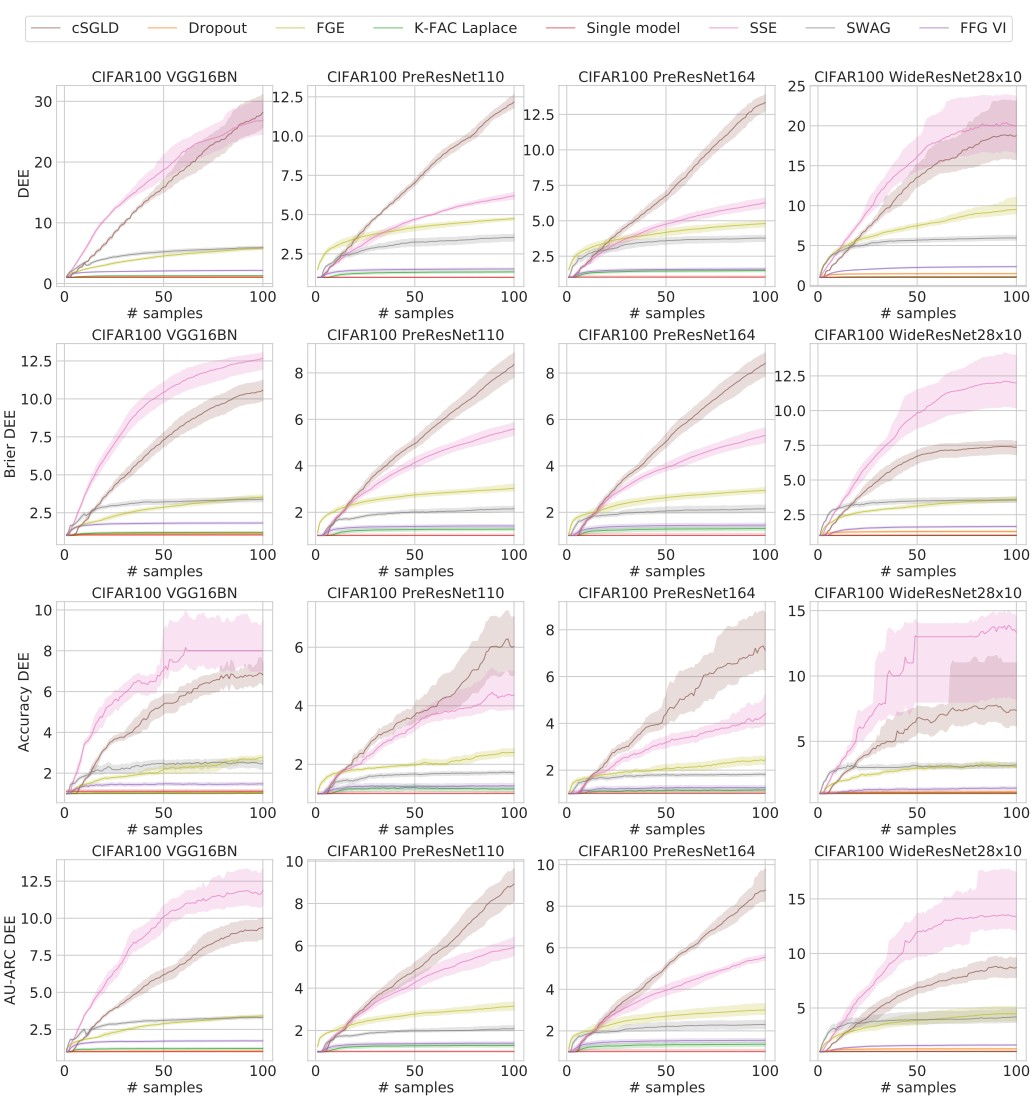

Figure 13: The deep ensemble equivalent of various ensembling techniques on CIFAR-100. Solid lines: mean DEE for different methods and architectures. Area between DEE$^{\text{lower}}$ and DEE$^{\text{upper}}$ is shaded. Lines 2–4 correspond to DEE based on other metrics, defined similarly to the log-likelihood-based DEE. Note that while the actual scale of DEE varies from metric to metric, the ordering of different methods and the overall behaviour of the lines remain the same.

| Model | Method | Error (%) | | | | Negative calibrated log-likelihood | | | |
|---|---|---|---|---|---|---|---|---|---|
| | | 1 | 5 | 10 | 100 | 1 | 5 | 10 | 100 |
| VGG16 CIFAR-10 | Dropout | $5.86_{\pm0.09}$ | $5.81_{\pm0.08}$ | $5.82_{\pm0.06}$ | $5.79_{\pm0.07}$ | $0.232_{\pm0.005}$ | $0.225_{\pm0.004}$ | $0.224_{\pm0.004}$ | $0.223_{\pm0.003}$ |
| | SWA-Gaussian | $7.03_{\pm0.50}$ | $5.66_{\pm0.08}$ | $5.49_{\pm0.12}$ | $5.25_{\pm0.13}$ | $0.230_{\pm0.014}$ | $0.182_{\pm0.003}$ | $0.171_{\pm0.002}$ | $0.160_{\pm0.002}$ |
| | Cyclic SGLD | $7.37_{\pm0.16}$ | $6.56_{\pm0.09}$ | $5.71_{\pm0.06}$ | $4.84_{\pm0.04}$ | $0.234_{\pm0.004}$ | $0.196_{\pm0.004}$ | $0.176_{\pm0.003}$ | $0.147_{\pm0.003}$ |
| | Fast Geometric Ens. | $6.52_{\pm0.16}$ | $5.95_{\pm0.16}$ | $5.69_{\pm0.16}$ | $5.10_{\pm0.13}$ | $0.213_{\pm0.005}$ | $0.187_{\pm0.003}$ | $0.178_{\pm0.003}$ | $0.155_{\pm0.004}$ |
| | Deep Ensembles | $5.95_{\pm0.14}$ | $4.79_{\pm0.11}$ | $4.57_{\pm0.07}$ | $4.39_{\pm\text{NA}}$ | $0.226_{\pm0.001}$ | $0.158_{\pm0.002}$ | $0.148_{\pm0.001}$ | $0.134_{\pm\text{NA}}$ |
| | Single model | $5.83_{\pm0.11}$ | $5.83_{\pm0.11}$ | $5.83_{\pm0.11}$ | $5.83_{\pm0.11}$ | $0.223_{\pm0.002}$ | $0.223_{\pm0.002}$ | $0.223_{\pm0.002}$ | $0.223_{\pm0.002}$ |
| | Variational Inf. (FFG) | $6.57_{\pm0.09}$ | $5.63_{\pm0.13}$ | $5.50_{\pm0.10}$ | $5.46_{\pm0.03}$ | $0.239_{\pm0.002}$ | $0.192_{\pm0.002}$ | $0.184_{\pm0.002}$ | $0.175_{\pm0.001}$ |
| | KFAC-Laplace | $6.00_{\pm0.13}$ | $5.82_{\pm0.12}$ | $5.82_{\pm0.19}$ | $5.80_{\pm0.19}$ | $0.210_{\pm0.005}$ | $0.203_{\pm0.007}$ | $0.201_{\pm0.007}$ | $0.200_{\pm0.008}$ |
| | Snapshot Ensembles | $7.76_{\pm0.22}$ | $5.52_{\pm0.13}$ | $5.00_{\pm0.10}$ | $4.54_{\pm0.05}$ | $0.247_{\pm0.005}$ | $0.176_{\pm0.001}$ | $0.160_{\pm0.001}$ | $0.137_{\pm0.001}$ |
| ResNet110 CIFAR-10 | SWA-Gaussian | $5.77_{\pm0.45}$ | $4.56_{\pm0.17}$ | $4.46_{\pm0.12}$ | $4.34_{\pm0.13}$ | $0.178_{\pm0.009}$ | $0.143_{\pm0.004}$ | $0.139_{\pm0.003}$ | $0.131_{\pm0.003}$ |
| | Cyclic SGLD | $6.18_{\pm0.20}$ | $5.32_{\pm0.15}$ | $4.55_{\pm0.13}$ | $3.83_{\pm0.02}$ | $0.185_{\pm0.006}$ | $0.156_{\pm0.005}$ | $0.138_{\pm0.002}$ | $0.115_{\pm0.001}$ |
| | Fast Geometric Ens. | $5.52_{\pm0.09}$ | $4.83_{\pm0.08}$ | $4.73_{\pm0.10}$ | $4.28_{\pm0.05}$ | $0.163_{\pm0.002}$ | $0.141_{\pm0.003}$ | $0.137_{\pm0.003}$ | $0.126_{\pm0.002}$ |
| | Deep Ensembles | $4.66_{\pm0.11}$ | $3.77_{\pm0.11}$ | $3.63_{\pm0.07}$ | $3.53_{\pm\text{NA}}$ | $0.148_{\pm0.004}$ | $0.117_{\pm0.002}$ | $0.112_{\pm0.002}$ | $0.106_{\pm\text{NA}}$ |
| | Single model | $4.69_{\pm0.11}$ | $4.69_{\pm0.11}$ | $4.69_{\pm0.11}$ | $4.69_{\pm0.11}$ | $0.150_{\pm0.002}$ | $0.150_{\pm0.002}$ | $0.150_{\pm0.002}$ | $0.150_{\pm0.002}$ |
| | Variational Inf. (FFG) | $5.57_{\pm0.26}$ | $4.91_{\pm0.15}$ | $4.72_{\pm0.13}$ | $4.60_{\pm0.03}$ | $0.178_{\pm0.003}$ | $0.149_{\pm0.001}$ | $0.144_{\pm0.001}$ | $0.140_{\pm0.000}$ |
| | KFAC-Laplace | $5.81_{\pm0.39}$ | $5.14_{\pm0.15}$ | $4.90_{\pm0.14}$ | $4.78_{\pm0.08}$ | $0.187_{\pm0.014}$ | $0.160_{\pm0.007}$ | $0.153_{\pm0.005}$ | $0.147_{\pm0.003}$ |
| | Snapshot Ensembles | $8.41_{\pm0.27}$ | $4.85_{\pm0.11}$ | $4.16_{\pm0.16}$ | $3.52_{\pm0.10}$ | $0.252_{\pm0.006}$ | $0.153_{\pm0.002}$ | $0.132_{\pm0.002}$ | $0.107_{\pm0.001}$ |
| ResNet164 CIFAR-10 | SWA-Gaussian | $5.41_{\pm0.71}$ | $4.21_{\pm0.19}$ | $4.21_{\pm0.23}$ | $4.02_{\pm0.14}$ | $0.171_{\pm0.028}$ | $0.130_{\pm0.004}$ | $0.128_{\pm0.004}$ | $0.121_{\pm0.002}$ |
| | Cyclic SGLD | $5.80_{\pm0.21}$ | $4.97_{\pm0.12}$ | $4.30_{\pm0.08}$ | $3.66_{\pm0.06}$ | $0.178_{\pm0.004}$ | $0.149_{\pm0.004}$ | $0.131_{\pm0.003}$ | $0.110_{\pm0.001}$ |
| | Fast Geometric Ens. | $5.22_{\pm0.07}$ | $4.49_{\pm0.06}$ | $4.36_{\pm0.07}$ | $4.09_{\pm0.12}$ | $0.157_{\pm0.003}$ | $0.134_{\pm0.002}$ | $0.130_{\pm0.001}$ | $0.119_{\pm0.002}$ |
| | Deep Ensembles | $4.53_{\pm0.11}$ | $3.51_{\pm0.09}$ | $3.50_{\pm0.06}$ | $3.34_{\pm\text{NA}}$ | $0.147_{\pm0.002}$ | $0.113_{\pm0.001}$ | $0.107_{\pm0.001}$ | $0.100_{\pm\text{NA}}$ |
| | Single model | $4.52_{\pm0.11}$ | $4.52_{\pm0.11}$ | $4.52_{\pm0.11}$ | $4.52_{\pm0.11}$ | $0.144_{\pm0.002}$ | $0.144_{\pm0.003}$ | $0.144_{\pm0.002}$ | $0.144_{\pm0.003}$ |
| | Variational Inf. (FFG) | $5.62_{\pm0.14}$ | $4.78_{\pm0.05}$ | $4.66_{\pm0.05}$ | $4.55_{\pm0.08}$ | $0.183_{\pm0.004}$ | $0.151_{\pm0.001}$ | $0.146_{\pm0.001}$ | $0.141_{\pm0.001}$ |
| | KFAC-Laplace | $5.23_{\pm0.29}$ | $4.77_{\pm0.23}$ | $4.65_{\pm0.17}$ | $4.60_{\pm0.09}$ | $0.168_{\pm0.004}$ | $0.151_{\pm0.007}$ | $0.146_{\pm0.005}$ | $0.142_{\pm0.004}$ |
| | Snapshot Ensembles | $8.06_{\pm0.10}$ | $4.50_{\pm0.04}$ | $3.89_{\pm0.09}$ | $3.50_{\pm0.05}$ | $0.241_{\pm0.004}$ | $0.144_{\pm0.003}$ | $0.124_{\pm0.002}$ | $0.104_{\pm0.001}$ |
| WideResNet CIFAR-10 | Dropout | $3.88_{\pm0.12}$ | $3.70_{\pm0.18}$ | $3.63_{\pm0.19}$ | $3.64_{\pm0.17}$ | $0.130_{\pm0.002}$ | $0.120_{\pm0.002}$ | $0.119_{\pm0.001}$ | $0.117_{\pm0.002}$ |
| | SWA-Gaussian | $4.98_{\pm1.17}$ | $3.53_{\pm0.09}$ | $3.34_{\pm0.14}$ | $3.28_{\pm0.10}$ | $0.157_{\pm0.036}$ | $0.111_{\pm0.004}$ | $0.105_{\pm0.003}$ | $0.101_{\pm0.002}$ |
| | Cyclic SGLD | $4.78_{\pm0.16}$ | $4.09_{\pm0.11}$ | $3.63_{\pm0.13}$ | $3.19_{\pm0.04}$ | $0.155_{\pm0.003}$ | $0.128_{\pm0.002}$ | $0.114_{\pm0.001}$ | $0.099_{\pm0.002}$ |
| | Fast Geometric Ens. | $4.86_{\pm0.17}$ | $3.95_{\pm0.07}$ | $3.77_{\pm0.10}$ | $3.34_{\pm0.06}$ | $0.148_{\pm0.003}$ | $0.120_{\pm0.002}$ | $0.113_{\pm0.002}$ | $0.102_{\pm0.001}$ |
| | Deep Ensembles | $3.65_{\pm0.02}$ | $3.11_{\pm0.10}$ | $3.01_{\pm0.06}$ | $2.83_{\pm\text{NA}}$ | $0.123_{\pm0.002}$ | $0.097_{\pm0.001}$ | $0.095_{\pm0.001}$ | $0.090_{\pm\text{NA}}$ |
| | Single model | $3.70_{\pm0.15}$ | $3.70_{\pm0.15}$ | $3.70_{\pm0.15}$ | $3.70_{\pm0.15}$ | $0.124_{\pm0.005}$ | $0.124_{\pm0.005}$ | $0.125_{\pm0.005}$ | $0.124_{\pm0.005}$ |
| | Variational Inf. (FFG) | $5.61_{\pm0.04}$ | $4.15_{\pm0.15}$ | $3.94_{\pm0.10}$ | $3.64_{\pm0.07}$ | $0.189_{\pm0.002}$ | $0.134_{\pm0.002}$ | $0.127_{\pm0.002}$ | $0.117_{\pm0.001}$ |
| | KFAC-Laplace | $4.03_{\pm0.19}$ | $3.90_{\pm0.15}$ | $3.88_{\pm0.22}$ | $3.83_{\pm0.16}$ | $0.134_{\pm0.004}$ | $0.124_{\pm0.004}$ | $0.122_{\pm0.005}$ | $0.120_{\pm0.003}$ |
| | Snapshot Ensembles | $5.56_{\pm0.15}$ | $3.68_{\pm0.09}$ | $3.33_{\pm0.10}$ | $2.89_{\pm0.07}$ | $0.179_{\pm0.005}$ | $0.119_{\pm0.001}$ | $0.105_{\pm0.001}$ | $0.090_{\pm0.001}$ |
| VGG16 CIFAR-100 | Dropout | $26.10_{\pm0.20}$ | $25.68_{\pm0.18}$ | $25.66_{\pm0.14}$ | $25.60_{\pm0.17}$ | $1.176_{\pm0.008}$ | $1.111_{\pm0.008}$ | $1.098_{\pm0.009}$ | $1.084_{\pm0.009}$ |
| | SWA-Gaussian | $27.74_{\pm1.87}$ | $24.53_{\pm0.09}$ | $23.64_{\pm0.28}$ | $22.97_{\pm0.20}$ | $1.109_{\pm0.073}$ | $0.931_{\pm0.007}$ | $0.879_{\pm0.007}$ | $0.826_{\pm0.005}$ |
| | Cyclic SGLD | $29.75_{\pm0.17}$ | $26.79_{\pm0.19}$ | $24.14_{\pm0.11}$ | $21.15_{\pm0.11}$ | $1.114_{\pm0.003}$ | $0.976_{\pm0.004}$ | $0.881_{\pm0.004}$ | $0.749_{\pm0.004}$ |
| | Fast Geometric Ens. | $27.07_{\pm0.24}$ | $25.35_{\pm0.29}$ | $24.68_{\pm0.40}$ | $22.78_{\pm0.22}$ | $1.057_{\pm0.010}$ | $0.965_{\pm0.003}$ | $0.930_{\pm0.003}$ | $0.827_{\pm0.004}$ |
| | Deep Ensembles | $25.72_{\pm0.17}$ | $21.60_{\pm0.13}$ | $20.79_{\pm0.16}$ | $19.88_{\pm\text{NA}}$ | $1.092_{\pm0.004}$ | $0.840_{\pm0.005}$ | $0.794_{\pm0.002}$ | $0.723_{\pm\text{NA}}$ |
| | Single model | $25.44_{\pm0.29}$ | $25.44_{\pm0.29}$ | $25.44_{\pm0.29}$ | $25.44_{\pm0.29}$ | $1.087_{\pm0.006}$ | $1.087_{\pm0.006}$ | $1.087_{\pm0.006}$ | $1.087_{\pm0.006}$ |
| | Variational Inf. (FFG) | $27.24_{\pm0.09}$ | $25.24_{\pm0.11}$ | $24.85_{\pm0.05}$ | $24.56_{\pm0.07}$ | $1.154_{\pm0.002}$ | $1.001_{\pm0.002}$ | $0.973_{\pm0.002}$ | $0.939_{\pm0.001}$ |
| | KFAC-Laplace | $27.11_{\pm0.59}$ | $25.98_{\pm0.21}$ | $25.84_{\pm0.38}$ | $25.70_{\pm0.38}$ | $1.174_{\pm0.037}$ | $1.089_{\pm0.007}$ | $1.069_{\pm0.005}$ | $1.050_{\pm0.008}$ |
| | Snapshot Ensembles | $31.19_{\pm0.33}$ | $23.87_{\pm0.18}$ | $22.31_{\pm0.31}$ | $21.03_{\pm0.10}$ | $1.170_{\pm0.012}$ | $0.899_{\pm0.004}$ | $0.834_{\pm0.005}$ | $0.751_{\pm0.003}$ |
| ResNet110 CIFAR-100 | SWA-Gaussian | $27.75_{\pm0.76}$ | $22.31_{\pm0.22}$ | $21.52_{\pm0.30}$ | $20.69_{\pm0.19}$ | $0.960_{\pm0.033}$ | $0.781_{\pm0.011}$ | $0.745_{\pm0.010}$ | $0.701_{\pm0.008}$ |
| | Cyclic SGLD | $25.73_{\pm0.14}$ | $23.30_{\pm0.19}$ | $21.20_{\pm0.21}$ | $18.07_{\pm0.16}$ | $0.914_{\pm0.006}$ | $0.818_{\pm0.004}$ | $0.753_{\pm0.002}$ | $0.630_{\pm0.002}$ |
| | Fast Geometric Ens. | $22.84_{\pm0.16}$ | $21.22_{\pm0.20}$ | $20.79_{\pm0.23}$ | $19.64_{\pm0.15}$ | $0.798_{\pm0.006}$ | $0.729_{\pm0.003}$ | $0.713_{\pm0.002}$ | $0.679_{\pm0.002}$ |
| | Deep Ensembles | $22.55_{\pm0.28}$ | $18.30_{\pm0.22}$ | $17.59_{\pm0.21}$ | $16.97_{\pm\text{NA}}$ | $0.847_{\pm0.007}$ | $0.675_{\pm0.001}$ | $0.638_{\pm0.001}$ | $0.594_{\pm\text{NA}}$ |
| | Single model | $22.66_{\pm0.31}$ | $22.66_{\pm0.31}$ | $22.66_{\pm0.31}$ | $22.66_{\pm0.31}$ | $0.848_{\pm0.014}$ | $0.848_{\pm0.015}$ | $0.848_{\pm0.014}$ | $0.848_{\pm0.015}$ |
| | Variational Inf. (FFG) | $24.27_{\pm0.26}$ | $22.41_{\pm0.13}$ | $22.14_{\pm0.12}$ | $21.86_{\pm0.07}$ | $0.924_{\pm0.007}$ | $0.829_{\pm0.003}$ | $0.813_{\pm0.001}$ | $0.795_{\pm0.001}$ |
| | KFAC-Laplace | $24.88_{\pm0.97}$ | $22.87_{\pm0.44}$ | $22.41_{\pm0.26}$ | $22.14_{\pm0.20}$ | $0.948_{\pm0.036}$ | $0.858_{\pm0.011}$ | $0.836_{\pm0.010}$ | $0.812_{\pm0.010}$ |
| | Snapshot Ensembles | $30.30_{\pm0.40}$ | $22.83_{\pm0.23}$ | $21.13_{\pm0.14}$ | $18.48_{\pm0.25}$ | $1.069_{\pm0.012}$ | $0.820_{\pm0.003}$ | $0.761_{\pm0.004}$ | $0.662_{\pm0.002}$ |
| ResNet164 CIFAR-100 | SWA-Gaussian | $24.38_{\pm0.93}$ | $20.62_{\pm0.18}$ | $20.08_{\pm0.19}$ | $19.48_{\pm0.19}$ | $0.844_{\pm0.042}$ | $0.719_{\pm0.006}$ | $0.700_{\pm0.006}$ | $0.667_{\pm0.004}$ |
| | Cyclic SGLD | $24.87_{\pm0.39}$ | $22.37_{\pm0.27}$ | $20.23_{\pm0.22}$ | $17.13_{\pm0.18}$ | $0.888_{\pm0.008}$ | $0.790_{\pm0.009}$ | $0.722_{\pm0.009}$ | $0.606_{\pm0.005}$ |
| | Fast Geometric Ens. | $21.92_{\pm0.15}$ | $20.10_{\pm0.22}$ | $19.87_{\pm0.25}$ | $18.73_{\pm0.25}$ | $0.765_{\pm0.003}$ | $0.699_{\pm0.004}$ | $0.686_{\pm0.004}$ | $0.650_{\pm0.003}$ |
| | Deep Ensembles | $21.41_{\pm0.25}$ | $17.53_{\pm0.17}$ | $16.90_{\pm0.15}$ | $16.50_{\pm\text{NA}}$ | $0.819_{\pm0.008}$ | $0.647_{\pm0.003}$ | $0.615_{\pm0.002}$ | $0.574_{\pm\text{NA}}$ |
| | Single model | $21.39_{\pm0.40}$ | $21.39_{\pm0.40}$ | $21.39_{\pm0.40}$ | $21.39_{\pm0.40}$ | $0.817_{\pm0.014}$ | $0.817_{\pm0.014}$ | $0.817_{\pm0.014}$ | $0.817_{\pm0.014}$ |
| | Variational Inf. (FFG) | $23.47_{\pm0.26}$ | $21.35_{\pm0.11}$ | $21.10_{\pm0.16}$ | $20.82_{\pm0.04}$ | $0.910_{\pm0.001}$ | $0.801_{\pm0.002}$ | $0.782_{\pm0.002}$ | $0.762_{\pm0.000}$ |
| | KFAC-Laplace | $23.44_{\pm0.45}$ | $21.77_{\pm0.20}$ | $21.29_{\pm0.23}$ | $21.03_{\pm0.38}$ | $0.902_{\pm0.019}$ | $0.813_{\pm0.006}$ | $0.792_{\pm0.005}$ | $0.772_{\pm0.007}$ |
| | Snapshot Ensembles | $29.48_{\pm0.19}$ | $21.92_{\pm0.18}$ | $20.27_{\pm0.23}$ | $17.68_{\pm0.07}$ | $1.045_{\pm0.005}$ | $0.789_{\pm0.005}$ | $0.729_{\pm0.004}$ | $0.634_{\pm0.003}$ |
| WideResNet CIFAR-100 | Dropout | $20.19_{\pm0.11}$ | $19.41_{\pm0.17}$ | $19.36_{\pm0.12}$ | $19.22_{\pm0.15}$ | $0.823_{\pm0.008}$ | $0.768_{\pm0.005}$ | $0.760_{\pm0.006}$ | $0.751_{\pm0.005}$ |
| | SWA-Gaussian | $20.45_{\pm0.73}$ | $17.57_{\pm0.17}$ | $17.21_{\pm0.22}$ | $17.08_{\pm0.19}$ | $0.794_{\pm0.025}$ | $0.653_{\pm0.004}$ | $0.634_{\pm0.005}$ | $0.614_{\pm0.005}$ |
| | Cyclic SGLD | $21.42_{\pm0.32}$ | $19.42_{\pm0.28}$ | $17.88_{\pm0.16}$ | $16.29_{\pm0.10}$ | $0.813_{\pm0.010}$ | $0.713_{\pm0.009}$ | $0.654_{\pm0.005}$ | $0.583_{\pm0.004}$ |
| | Fast Geometric Ens. | $21.48_{\pm0.31}$ | $18.54_{\pm0.16}$ | $18.00_{\pm0.19}$ | $17.12_{\pm0.16}$ | $0.770_{\pm0.007}$ | $0.652_{\pm0.006}$ | $0.630_{\pm0.006}$ | $0.596_{\pm0.003}$ |
| | Deep Ensembles | $19.38_{\pm0.20}$ | $16.55_{\pm0.08}$ | $16.17_{\pm0.15}$ | $15.77_{\pm\text{NA}}$ | $0.797_{\pm0.007}$ | $0.623_{\pm0.003}$ | $0.595_{\pm0.003}$ | $0.571_{\pm\text{NA}}$ |
| | Single model | $19.31_{\pm0.24}$ | $19.31_{\pm0.24}$ | $19.31_{\pm0.24}$ | $19.31_{\pm0.24}$ | $0.797_{\pm0.010}$ | $0.797_{\pm0.010}$ | $0.797_{\pm0.010}$ | $0.797_{\pm0.010}$ |
| | Variational Inf. (FFG) | $24.38_{\pm0.27}$ | $20.17_{\pm0.15}$ | $19.28_{\pm0.09}$ | $18.74_{\pm0.08}$ | $1.004_{\pm0.011}$ | $0.767_{\pm0.004}$ | $0.727_{\pm0.003}$ | $0.685_{\pm0.002}$ |
| | KFAC-Laplace | $20.02_{\pm0.18}$ | $19.76_{\pm0.15}$ | $19.53_{\pm0.19}$ | $19.43_{\pm0.21}$ | $0.834_{\pm0.009}$ | $0.803_{\pm0.006}$ | $0.795_{\pm0.007}$ | $0.789_{\pm0.006}$ |
| | Snapshot Ensembles | $23.01_{\pm0.26}$ | $18.20_{\pm0.13}$ | $17.12_{\pm0.31}$ | $16.07_{\pm0.07}$ | $0.859_{\pm0.009}$ | $0.678_{\pm0.006}$ | $0.633_{\pm0.008}$ | $0.582_{\pm0.004}$ |

Table 3: Classification error and negative calibrated log-likelihood for different models and numbers of samples on CIFAR-10/100.

| Model | Method | Error (%) | | | Negative calibrated log-likelihood | | |
|---|---|---|---|---|---|---|---|
| | | 5 | 10 | 100 | 5 | 10 | 100 |
| VGG16 CIFAR-10 | Dropout | 5.81 vs 5.52↓ | 5.82 vs 5.34↓ | 5.79 vs 5.21↓ | 0.225 vs 0.187↓ | 0.224 vs 0.177↓ | 0.223 vs 0.167↓ |
| | SWA-Gaussian | 5.66 vs 5.65≈ | 5.49 vs 5.36≈ | 5.25 vs 5.08↓ | 0.182 vs 0.180≈ | 0.171 vs 0.166↓ | 0.160 vs 0.152↓ |
| | Cyclic SGLD | 6.56 vs 6.07↓ | 5.71 vs 5.47↓ | 4.84 vs 4.88≈ | 0.196 vs 0.186↓ | 0.176 vs 0.169↓ | 0.147 vs 0.147≈ |
| | Fast Geometric Ens. | 5.95 vs 5.64↓ | 5.69 vs 5.36↓ | 5.10 vs 4.98≈ | 0.187 vs 0.177↓ | 0.178 vs 0.167↓ | 0.155 vs 0.150↓ |
| | Deep Ensembles | 4.79 vs 4.90≈ | 4.57 vs 4.73↑ | 4.39 vs 4.55↑ | 0.158 vs 0.162↑ | 0.148 vs 0.152↑ | 0.134 vs 0.136↑ |
| | Single model | 5.83 vs 5.55↓ | 5.83 vs 5.35↓ | 5.83 vs 5.19↓ | 0.223 vs 0.183↓ | 0.223 vs 0.174↓ | 0.223 vs 0.166↓ |
| | Variational Inf. (FFG) | 5.63 vs 5.43↓ | 5.50 vs 5.25↓ | 5.46 vs 5.07↓ | 0.192 vs 0.182↓ | 0.184 vs 0.169↓ | 0.175 vs 0.154↓ |
| | KFAC-Laplace | 5.82 vs 5.49↓ | 5.82 vs 5.32↓ | 5.80 vs 5.17↓ | 0.203 vs 0.177↓ | 0.201 vs 0.171↓ | 0.200 vs 0.164↓ |
| | Snapshot Ensembles | 5.52 vs 5.61≈ | 5.00 vs 5.03≈ | 4.54 vs 4.64↑ | 0.176 vs 0.178↑ | 0.160 vs 0.160≈ | 0.137 vs 0.137≈ |
| ResNet110 CIFAR-10 | SWA-Gaussian | 4.56 vs 4.47≈ | 4.46 vs 4.34↓ | 4.34 vs 4.16↓ | 0.143 vs 0.142≈ | 0.139 vs 0.135↓ | 0.131 vs 0.125↓ |
| | Cyclic SGLD | 5.32 vs 4.88↓ | 4.55 vs 4.35↓ | 3.83 vs 3.59↓ | 0.156 vs 0.148↓ | 0.138 vs 0.131↓ | 0.115 vs 0.111↓ |
| | Fast Geometric Ens. | 4.83 vs 4.59↓ | 4.73 vs 4.44↓ | 4.28 vs 4.14↓ | 0.141 vs 0.138≈ | 0.137 vs 0.132↓ | 0.126 vs 0.121↓ |
| | Deep Ensembles | 3.77 vs 3.70↓ | 3.63 vs 3.58≈ | 3.53 vs 3.45↓ | 0.117 vs 0.118≈ | 0.112 vs 0.112≈ | 0.106 vs 0.105↓ |
| | Single model | 4.69 vs 4.32↓ | 4.69 vs 4.23↓ | 4.69 vs 4.10↓ | 0.150 vs 0.137↓ | 0.150 vs 0.134↓ | 0.150 vs 0.131↓ |
| | Variational Inf. (FFG) | 4.91 vs 4.66↓ | 4.72 vs 4.43↓ | 4.60 vs 4.25↓ | 0.149 vs 0.145↓ | 0.144 vs 0.138↓ | 0.140 vs 0.130↓ |
| | KFAC-Laplace | 5.14 vs 4.43↓ | 4.90 vs 4.27↓ | 4.78 vs 4.17↓ | 0.160 vs 0.139↓ | 0.153 vs 0.134↓ | 0.147 vs 0.130↓ |
| | Snapshot Ensembles | 4.85 vs 4.78↓ | 4.16 vs 4.18≈ | 3.52 vs 3.48≈ | 0.153 vs 0.150↓ | 0.132 vs 0.130↓ | 0.107 vs 0.106↓ |
| ResNet164 CIFAR-10 | SWA-Gaussian | 4.21 vs 4.17≈ | 4.21 vs 4.04≈ | 4.02 vs 3.84↓ | 0.130 vs 0.130≈ | 0.128 vs 0.126↓ | 0.121 vs 0.116↓ |
| | Cyclic SGLD | 4.97 vs 4.67↓ | 4.30 vs 4.01↓ | 3.66 vs 3.48↓ | 0.149 vs 0.141↓ | 0.131 vs 0.125↓ | 0.110 vs 0.107↓ |
| | Fast Geometric Ens. | 4.49 vs 4.29↓ | 4.36 vs 4.15↓ | 4.09 vs 3.85↓ | 0.134 vs 0.131↓ | 0.130 vs 0.126↓ | 0.119 vs 0.115↓ |
| | Deep Ensembles | 3.51 vs 3.58↑ | 3.50 vs 3.46≈ | 3.34 vs 3.30↓ | 0.113 vs 0.114↑ | 0.107 vs 0.107≈ | 0.100 vs 0.101↑ |
| | Single model | 4.52 vs 4.15↓ | 4.52 vs 4.08↓ | 4.52 vs 3.98↓ | 0.144 vs 0.131↓ | 0.144 vs 0.128↓ | 0.144 vs 0.126↓ |
| | Variational Inf. (FFG) | 4.78 vs 4.41↓ | 4.66 vs 4.22↓ | 4.55 vs 4.05↓ | 0.151 vs 0.142↓ | 0.146 vs 0.135↓ | 0.141 vs 0.128↓ |
| | KFAC-Laplace | 4.77 vs 4.26↓ | 4.65 vs 4.21↓ | 4.60 vs 4.08↓ | 0.151 vs 0.135↓ | 0.146 vs 0.132↓ | 0.142 vs 0.127↓ |
| | Snapshot Ensembles | 4.50 vs 4.42↓ | 3.89 vs 3.87≈ | 3.50 vs 3.35↓ | 0.144 vs 0.142↓ | 0.124 vs 0.123↓ | 0.104 vs 0.104≈ |
| WideResNet CIFAR-10 | Dropout | 3.70 vs 3.62≈ | 3.63 vs 3.52↓ | 3.64 vs 3.44↓ | 0.120 vs 0.117↓ | 0.119 vs 0.114↓ | 0.117 vs 0.111↓ |
| | SWA-Gaussian | 3.53 vs 3.59≈ | 3.34 vs 3.34≈ | 3.28 vs 3.24≈ | 0.111 vs 0.114≈ | 0.105 vs 0.107↑ | 0.101 vs 0.102↑ |
| | Cyclic SGLD | 4.09 vs 3.95≈ | 3.63 vs 3.58≈ | 3.19 vs 3.22≈ | 0.128 vs 0.125↓ | 0.114 vs 0.112↓ | 0.099 vs 0.100↑ |
| | Fast Geometric Ens. | 3.95 vs 3.90≈ | 3.77 vs 3.64↓ | 3.34 vs 3.27↓ | 0.120 vs 0.120≈ | 0.113 vs 0.113≈ | 0.102 vs 0.102≈ |
| | Deep Ensembles | 3.11 vs 3.21↑ | 3.01 vs 3.02≈ | 2.83 vs 2.91↑ | 0.097 vs 0.103↑ | 0.095 vs 0.098↑ | 0.090 vs 0.094↑ |
| | Single model | 3.70 vs 3.53↓ | 3.70 vs 3.45↓ | 3.70 vs 3.40↓ | 0.124 vs 0.117↓ | 0.125 vs 0.114↓ | 0.124 vs 0.113↓ |
| | Variational Inf. (FFG) | 4.15 vs 4.05≈ | 3.94 vs 3.80↓ | 3.64 vs 3.63≈ | 0.134 vs 0.136≈ | 0.127 vs 0.126≈ | 0.117 vs 0.116↓ |
| | KFAC-Laplace | 3.90 vs 3.63↓ | 3.88 vs 3.58↓ | 3.83 vs 3.50↓ | 0.124 vs 0.115↓ | 0.122 vs 0.113↓ | 0.120 vs 0.111↓ |
| | Snapshot Ensembles | 3.68 vs 3.74≈ | 3.33 vs 3.35≈ | 2.89 vs 2.90≈ | 0.119 vs 0.122↑ | 0.105 vs 0.107↑ | 0.090 vs 0.093↑ |
| VGG16 CIFAR-100 | Dropout | 25.68 vs 24.37↓ | 25.66 vs 23.89↓ | 25.60 vs 23.41↓ | 1.111 vs 0.999↓ | 1.098 vs 0.960↓ | 1.084 vs 0.911↓ |
| | SWA-Gaussian | 24.53 vs 24.28≈ | 23.64 vs 23.27↓ | 22.97 vs 22.34↓ | 0.931 vs 0.926≈ | 0.879 vs 0.859↓ | 0.826 vs 0.795↓ |
| | Cyclic SGLD | 26.79 vs 25.66↓ | 24.14 vs 23.45↓ | 21.15 vs 21.04≈ | 0.976 vs 0.929↓ | 0.881 vs 0.848↓ | 0.749 vs 0.740↓ |
| | Fast Geometric Ens. | 25.35 vs 24.53↓ | 24.68 vs 23.62↓ | 22.78 vs 22.20↓ | 0.965 vs 0.921↓ | 0.930 vs 0.878↓ | 0.827 vs 0.800↓ |
| | Deep Ensembles | 21.60 vs 21.90↑ | 20.79 vs 21.03↑ | 19.88 vs 20.23↑ | 0.840 vs 0.865↑ | 0.794 vs 0.811↑ | 0.723 vs 0.731↑ |
| | Single model | 25.44 vs 24.38↓ | 25.44 vs 23.92↓ | 25.44 vs 23.48↓ | 1.087 vs 0.973↓ | 1.087 vs 0.945↓ | 1.087 vs 0.912↓ |
| | Variational Inf. (FFG) | 25.24 vs 24.19↓ | 24.85 vs 23.56↓ | 24.56 vs 22.89↓ | 1.001 vs 0.964↓ | 0.973 vs 0.919↓ | 0.939 vs 0.864↓ |
| | KFAC-Laplace | 25.98 vs 24.53↓ | 25.84 vs 24.03↓ | 25.70 vs 23.57↓ | 1.089 vs 0.989↓ | 1.069 vs 0.949↓ | 1.050 vs 0.909↓ |
| | Snapshot Ensembles | 23.87 vs 23.88≈ | 22.31 vs 22.50≈ | 21.03 vs 21.04≈ | 0.899 vs 0.905≈ | 0.834 vs 0.836≈ | 0.751 vs 0.750≈ |
| ResNet110 CIFAR-100 | SWA-Gaussian | 22.31 vs 22.20≈ | 21.52 vs 21.21≈ | 20.69 vs 20.28↓ | 0.781 vs 0.770↓ | 0.745 vs 0.730↓ | 0.701 vs 0.683↓ |
| | Cyclic SGLD | 23.30 vs 22.32↓ | 21.20 vs 20.43↓ | 18.07 vs 17.67↓ | 0.818 vs 0.787↓ | 0.753 vs 0.725↓ | 0.630 vs 0.616↓ |
| | Fast Geometric Ens. | 21.22 vs 20.69↓ | 20.79 vs 20.18↓ | 19.64 vs 19.25↓ | 0.729 vs 0.714↓ | 0.713 vs 0.694↓ | 0.679 vs 0.661↓ |
| | Deep Ensembles | 18.30 vs 18.36≈ | 17.59 vs 17.61≈ | 16.97 vs 16.74↓ | 0.675 vs 0.672≈ | 0.638 vs 0.635↓ | 0.594 vs 0.591↓ |
| | Single model | 22.66 vs 21.37↓ | 22.66 vs 21.17↓ | 22.66 vs 20.98↓ | 0.848 vs 0.797↓ | 0.848 vs 0.786↓ | 0.848 vs 0.775↓ |
| | Variational Inf. (FFG) | 22.41 vs 21.67↓ | 22.14 vs 21.21↓ | 21.86 vs 20.77↓ | 0.829 vs 0.799↓ | 0.813 vs 0.775↓ | 0.795 vs 0.748↓ |
| | KFAC-Laplace | 22.87 vs 21.69↓ | 22.41 vs 21.28↓ | 22.14 vs 20.99↓ | 0.858 vs 0.810↓ | 0.836 vs 0.788↓ | 0.812 vs 0.766↓ |
| | Snapshot Ensembles | 22.83 vs 22.33↓ | 21.13 vs 20.71↓ | 18.48 vs 18.23≈ | 0.820 vs 0.806↓ | 0.761 vs 0.744↓ | 0.662 vs 0.651↓ |
| ResNet164 CIFAR-100 | SWA-Gaussian | 20.62 vs 20.61≈ | 20.08 vs 20.08≈ | 19.48 vs 19.33↓ | 0.719 vs 0.715≈ | 0.700 vs 0.690↓ | 0.667 vs 0.654↓ |
| | Cyclic SGLD | 22.37 vs 21.57↓ | 20.23 vs 19.58↓ | 17.13 vs 16.99↓ | 0.790 vs 0.767↓ | 0.722 vs 0.702↓ | 0.606 vs 0.595↓ |
| | Fast Geometric Ens. | 20.10 vs 19.82↓ | 19.87 vs 19.39↓ | 18.73 vs 18.33↓ | 0.699 vs 0.689↓ | 0.686 vs 0.671↓ | 0.650 vs 0.636↓ |
| | Deep Ensembles | 17.53 vs 17.57≈ | 16.90 vs 16.84≈ | 16.50 vs 16.22↓ | 0.647 vs 0.650↑ | 0.615 vs 0.613≈ | 0.574 vs 0.575↑ |
| | Single model | 21.39 vs 20.60↓ | 21.39 vs 20.39↓ | 21.39 vs 20.23↓ | 0.817 vs 0.772↓ | 0.817 vs 0.761↓ | 0.817 vs 0.751↓ |
| | Variational Inf. (FFG) | 21.35 vs 21.06↓ | 21.10 vs 20.54↓ | 20.82 vs 19.97↓ | 0.801 vs 0.785↓ | 0.782 vs 0.759↓ | 0.762 vs 0.731↓ |
| | KFAC-Laplace | 21.77 vs 20.66↓ | 21.29 vs 20.36↓ | 21.03 vs 20.18↓ | 0.813 vs 0.778↓ | 0.792 vs 0.758↓ | 0.772 vs 0.740↓ |
| | Snapshot Ensembles | 21.92 vs 21.69↓ | 20.27 vs 19.92↓ | 17.68 vs 17.66≈ | 0.789 vs 0.781↓ | 0.729 vs 0.720↓ | 0.634 vs 0.629↓ |
| WideResNet CIFAR-100 | Dropout | 19.41 vs 19.27≈ | 19.36 vs 19.11↓ | 19.22 vs 18.88↓ | 0.768 vs 0.751↓ | 0.760 vs 0.738↓ | 0.751 vs 0.723↓ |
| | SWA-Gaussian | 17.57 vs 17.79↑ | 17.21 vs 17.27≈ | 17.08 vs 16.89↓ | 0.653 vs 0.658↑ | 0.634 vs 0.635≈ | 0.614 vs 0.610≈ |
| | Cyclic SGLD | 19.42 vs 18.83↓ | 17.88 vs 17.50↓ | 16.29 vs 16.21≈ | 0.713 vs 0.696↓ | 0.654 vs 0.641↓ | 0.583 vs 0.580↓ |
| | Fast Geometric Ens. | 18.54 vs 18.39↓ | 18.00 vs 17.84↓ | 17.12 vs 16.93↓ | 0.652 vs 0.649↓ | 0.630 vs 0.624↓ | 0.596 vs 0.592↓ |
| | Deep Ensembles | 16.55 vs 16.84↑ | 16.17 vs 16.30≈ | 15.77 vs 15.77≈ | 0.623 vs 0.632↑ | 0.595 vs 0.602↑ | 0.571 vs 0.573↑ |
| | Single model | 19.31 vs 18.83↓ | 19.31 vs 18.80↓ | 19.31 vs 18.72↓ | 0.797 vs 0.755↓ | 0.797 vs 0.746↓ | 0.797 vs 0.738↓ |
| | Variational Inf. (FFG) | 20.17 vs 20.12≈ | 19.28 vs 19.20↓ | 18.74 vs 18.54↓ | 0.767 vs 0.766≈ | 0.727 vs 0.724↓ | 0.685 vs 0.679↓ |
| | KFAC-Laplace | 19.76 vs 19.21↓ | 19.53 vs 19.03↓ | 19.43 vs 18.93↓ | 0.803 vs 0.764↓ | 0.795 vs 0.757↓ | 0.789 vs 0.747↓ |
| | Snapshot Ensembles | 18.20 vs 18.22≈ | 17.12 vs 17.20≈ | 16.07 vs 16.27≈ | 0.678 vs 0.680≈ | 0.633 vs 0.635≈ | 0.582 vs 0.586↑ |

Table 4: Classification error and negative calibrated log-likelihood before vs. after test-time augmentation on CIFAR-10/100.

| Model | Method | Deep ensemble equivalent score | | | | |
|---|---|---|---|---|---|---|
| | | 1 sample | 5 samples | 10 samples | 50 samples | 100 samples |
| VGG16 CIFAR-10 | Deep Ensembles | 1.0 | 5.0 | 10.0 | 50.0 | 100.0 |
| | Snapshot Ensembles | 1.0 | 2.6 | 4.7 | 21.9 | 33.8 |
| | Cyclic SGLD | 1.0 | 1.7 | 2.6 | 7.4 | 10.5 |
| | SWA-Gaussian | 1.0 | 2.2 | 2.9 | 4.4 | 4.6 |
| | Fast Geometric Ens. | 1.3 | 2.0 | 2.4 | 4.7 | 5.8 |
| | Dropout | 1.0 | 1.0 | 1.0 | 1.1 | 1.1 |
| | Variational Inf. (FFG) | 1.0 | 1.8 | 2.0 | 2.5 | 2.6 |
| | KFAC-Laplace | 1.4 | 1.6 | 1.6 | 1.6 | 1.6 |
| | Single model | 1.1 | 1.1 | 1.1 | 1.1 | 1.1 |
| ResNet110 CIFAR-10 | Deep Ensembles | 1.0 | 5.0 | 10.0 | 48.0 | 94.6 |
| | Snapshot Ensembles | 1.0 | 1.0 | 2.0 | 14.6 | 32.6 |
| | Cyclic SGLD | 1.0 | 1.0 | 1.6 | 4.6 | 6.0 |
| | SWA-Gaussian | 1.0 | 1.3 | 1.5 | 1.9 | 2.0 |
| | Fast Geometric Ens. | 1.0 | 1.4 | 1.7 | 2.4 | 2.9 |
| | Variational Inf. (FFG) | 1.0 | 1.0 | 1.2 | 1.4 | 1.5 |
| | KFAC-Laplace | 1.0 | 1.0 | 1.0 | 1.0 | 1.1 |
| | Single model | 1.0 | 1.0 | 1.0 | 1.0 | 1.0 |
| ResNet164 CIFAR-10 | Deep Ensembles | 1.0 | 5.0 | 10.0 | 43.7 | 100.0 |
| | Snapshot Ensembles | 1.0 | 1.1 | 2.3 | 11.3 | 17.2 |
| | Cyclic SGLD | 1.0 | 1.0 | 1.8 | 4.8 | 6.7 |
| | SWA-Gaussian | 1.0 | 1.8 | 1.9 | 2.5 | 2.7 |
| | Fast Geometric Ens. | 1.0 | 1.6 | 1.8 | 2.5 | 2.9 |
| | Variational Inf. (FFG) | 1.0 | 1.0 | 1.0 | 1.3 | 1.3 |
| | KFAC-Laplace | 1.0 | 1.0 | 1.0 | 1.2 | 1.2 |
| | Single model | 1.1 | 1.1 | 1.1 | 1.1 | 1.1 |
| WideResNet CIFAR-10 | Deep Ensembles | 1.0 | 5.0 | 10.0 | 48.0 | 86.0 |
| | Snapshot Ensembles | 1.0 | 1.3 | 2.5 | 30.6 | 100.0 |
| | Cyclic SGLD | 1.0 | 1.0 | 1.6 | 4.0 | 4.1 |
| | SWA-Gaussian | 1.0 | 1.8 | 2.5 | 3.1 | 3.3 |
| | Fast Geometric Ens. | 1.0 | 1.2 | 1.7 | 2.7 | 3.2 |
| | Dropout | 1.0 | 1.2 | 1.3 | 1.4 | 1.4 |
| | Variational Inf. (FFG) | 1.0 | 1.0 | 1.0 | 1.3 | 1.4 |
| | KFAC-Laplace | 1.0 | 1.0 | 1.0 | 1.2 | 1.2 |
| | Single model | 1.0 | 1.0 | 1.0 | 1.0 | 1.0 |
| VGG16 CIFAR-100 | Deep Ensembles | 1.0 | 5.0 | 10.0 | 50.0 | 100.0 |
| | Snapshot Ensembles | 1.0 | 2.9 | 5.4 | 18.8 | 26.8 |
| | Cyclic SGLD | 1.0 | 1.8 | 3.3 | 15.9 | 28.2 |
| | SWA-Gaussian | 1.0 | 2.3 | 3.4 | 5.2 | 5.9 |
| | Fast Geometric Ens. | 1.2 | 1.9 | 2.3 | 4.5 | 5.8 |
| | Dropout | 1.0 | 1.0 | 1.0 | 1.0 | 1.1 |
| | Variational Inf. (FFG) | 1.0 | 1.6 | 1.8 | 2.1 | 2.2 |
| | KFAC-Laplace | 1.0 | 1.0 | 1.2 | 1.3 | 1.3 |
| | Single model | 1.0 | 1.0 | 1.0 | 1.0 | 1.0 |
| ResNet110 CIFAR-100 | Deep Ensembles | 1.0 | 5.0 | 10.0 | 50.0 | 99.0 |
| | Snapshot Ensembles | 1.0 | 1.3 | 1.9 | 4.7 | 6.2 |
| | Cyclic SGLD | 1.0 | 1.3 | 2.0 | 7.0 | 12.2 |
| | SWA-Gaussian | 1.0 | 1.7 | 2.2 | 3.3 | 3.6 |
| | Fast Geometric Ens. | 1.5 | 2.6 | 3.0 | 4.2 | 4.7 |
| | Variational Inf. (FFG) | 1.0 | 1.2 | 1.3 | 1.5 | 1.5 |
| | KFAC-Laplace | 1.0 | 1.0 | 1.1 | 1.3 | 1.4 |
| | Single model | 1.0 | 1.0 | 1.0 | 1.0 | 1.0 |
| ResNet164 CIFAR-100 | Deep Ensembles | 1.0 | 5.0 | 10.0 | 50.0 | 100.0 |
| | Snapshot Ensembles | 1.0 | 1.3 | 1.9 | 4.8 | 6.3 |
| | Cyclic SGLD | 1.0 | 1.3 | 2.0 | 6.8 | 13.3 |
| | SWA-Gaussian | 1.0 | 2.1 | 2.6 | 3.6 | 3.8 |
| | Fast Geometric Ens. | 1.6 | 2.6 | 3.0 | 4.2 | 4.8 |
| | Variational Inf. (FFG) | 1.0 | 1.2 | 1.4 | 1.6 | 1.6 |
| | KFAC-Laplace | 1.0 | 1.1 | 1.3 | 1.5 | 1.5 |
| | Single model | 1.0 | 1.0 | 1.0 | 1.0 | 1.0 |
| WideResNet CIFAR-100 | Deep Ensembles | 1.0 | 5.0 | 10.0 | 49.0 | 95.9 |
| | Snapshot Ensembles | 1.0 | 2.5 | 4.2 | 16.2 | 20.0 |
| | Cyclic SGLD | 1.0 | 1.9 | 3.2 | 13.6 | 18.8 |
| | SWA-Gaussian | 1.0 | 3.2 | 4.2 | 5.7 | 6.0 |
| | Fast Geometric Ens. | 1.3 | 3.2 | 4.4 | 7.5 | 9.5 |
| | Dropout | 1.0 | 1.3 | 1.4 | 1.5 | 1.5 |
| | Variational Inf. (FFG) | 1.0 | 1.3 | 1.7 | 2.2 | 2.4 |
| | KFAC-Laplace | 1.0 | 1.0 | 1.0 | 1.1 | 1.1 |
| | Single model | 1.0 | 1.0 | 1.0 | 1.0 | 1.0 |

Table 5: Deep ensemble equivalent score for CIFAR-10/100.

| Model | Method | Error (%) | | | | Negative calibrated log-likelihood | | | |
|---|---|---|---|---|---|---|---|---|---|
| | | 1 | 5 | 10 | 50 | 1 | 5 | 10 | 50 |
| | Fast Geometric Ens. | $23.71_{\pm0.00}$ | $23.61_{\pm0.00}$ | $23.56_{\pm0.00}$ | $23.28_{\pm0.00}$ | $0.929_{\pm0.000}$ | $0.921_{\pm0.000}$ | $0.916_{\pm0.000}$ | $0.904_{\pm0.000}$ |
| | Deep Ensembles | $23.79_{\pm0.14}$ | $21.19_{\pm0.14}$ | $20.90_{\pm0.08}$ | $20.63_{\pm\text{NA}}$ | $0.935_{\pm0.007}$ | $0.823_{\pm0.002}$ | $0.805_{\pm0.000}$ | $0.788_{\pm\text{NA}}$ |
| ResNet50 | Single model | $23.86_{\pm0.20}$ | $23.86_{\pm0.20}$ | $23.86_{\pm0.20}$ | $23.86_{\pm0.20}$ | $0.938_{\pm0.006}$ | $0.938_{\pm0.006}$ | $0.938_{\pm0.006}$ | $0.938_{\pm0.006}$ |
| | Variational Inf. (FFG) | $24.50_{\pm0.06}$ | $23.82_{\pm0.03}$ | $23.77_{\pm0.04}$ | $23.67_{\pm0.00}$ | $0.957_{\pm0.001}$ | $0.927_{\pm0.000}$ | $0.923_{\pm0.001}$ | $0.920_{\pm0.000}$ |
| | KFAC-Laplace | $25.01_{\pm0.49}$ | $24.19_{\pm0.29}$ | $23.93_{\pm0.20}$ | $23.86_{\pm0.16}$ | $0.988_{\pm0.022}$ | $0.948_{\pm0.013}$ | $0.939_{\pm0.011}$ | $0.934_{\pm0.008}$ |
| | Snapshot Ensembles | $24.92_{\pm\text{NA}}$ | $22.21_{\pm\text{NA}}$ | $21.75_{\pm\text{NA}}$ | $21.48_{\pm\text{NA}}$ | $0.983_{\pm\text{NA}}$ | $0.865_{\pm\text{NA}}$ | $0.843_{\pm\text{NA}}$ | $0.830_{\pm\text{NA}}$ |

Table 6: Classification error and negative calibrated log-likelihood for different numbers of samples on ImageNet.

| Model | Method | Error (%) | | | Negative calibrated log-likelihood | | |
|---|---|---|---|---|---|---|---|
| | | 5 | 10 | 50 | 5 | 10 | 50 |
| | Fast Geometric Ens. | 23.61 vs 22.21$_\downarrow$ | 23.56 vs 21.37$_\downarrow$ | 23.28 vs 20.67$_\downarrow$ | 0.921 vs 0.894$_\downarrow$ | 0.916 vs 0.842$_\downarrow$ | 0.904 vs 0.793$_\downarrow$ |
| | Deep Ensembles | 21.19 vs 21.20$_\approx$ | 20.90 vs 20.16$_\downarrow$ | 20.63 vs 19.39$_\downarrow$ | 0.823 vs 0.855$_\uparrow$ | 0.805 vs 0.793$_\downarrow$ | 0.788 vs 0.739$_\downarrow$ |
| ResNet50 | Single model | 23.86 vs 22.39$_\downarrow$ | 23.86 vs 21.60$_\downarrow$ | 23.86 vs 21.06$_\downarrow$ | 0.938 vs 0.900$_\downarrow$ | 0.938 vs 0.851$_\downarrow$ | 0.938 vs 0.805$_\downarrow$ |
| | Variational Inf. (FFG) | 23.82 vs 22.58$_\downarrow$ | 23.77 vs 21.74$_\downarrow$ | 23.67 vs 21.10$_\downarrow$ | 0.927 vs 0.905$_\downarrow$ | 0.923 vs 0.851$_\downarrow$ | 0.920 vs 0.805$_\downarrow$ |
| | KFAC-Laplace | 24.19 vs 22.50$_\downarrow$ | 23.93 vs 21.67$_\downarrow$ | 23.86 vs 21.04$_\downarrow$ | 0.948 vs 0.906$_\downarrow$ | 0.939 vs 0.855$_\downarrow$ | 0.934 vs 0.809$_\downarrow$ |
| | Snapshot Ensembles | 22.21 vs 21.99$_\downarrow$ | 21.75 vs 20.81$_\downarrow$ | 21.48 vs 19.86$_\downarrow$ | 0.865 vs 0.879$_\uparrow$ | 0.843 vs 0.815$_\downarrow$ | 0.830 vs 0.763$_\downarrow$ |

Table 7: Classification error and negative calibrated log-likelihood before vs. after test-time augmentation on ImageNet.

| Model | Method | Deep ensemble equivalent score | | | | |
|---|---|---|---|---|---|---|
| | | 1 sample | 5 samples | 10 samples | 25 samples | 50 samples |
| | Deep Ensembles | 1.0 | 5.0 | 10.0 | 25.0 | 50.0 |
| | Snapshot Ensembles | 1.0 | 2.2 | 3.2 | 4.0 | 4.2 |
| ResNet50 | Fast Geometric Ens. | 1.1 | 1.2 | 1.3 | 1.4 | 1.5 |
| | Variational Inf. (FFG) | 1.0 | 1.1 | 1.2 | 1.2 | 1.2 |
| | KFAC-Laplace | 1.0 | 1.0 | 1.0 | 1.0 | 1.0 |
| | Single model | 1.0 | 1.0 | 1.0 | 1.0 | 1.0 |

Table 8: Deep ensemble equivalent score for ImageNet.

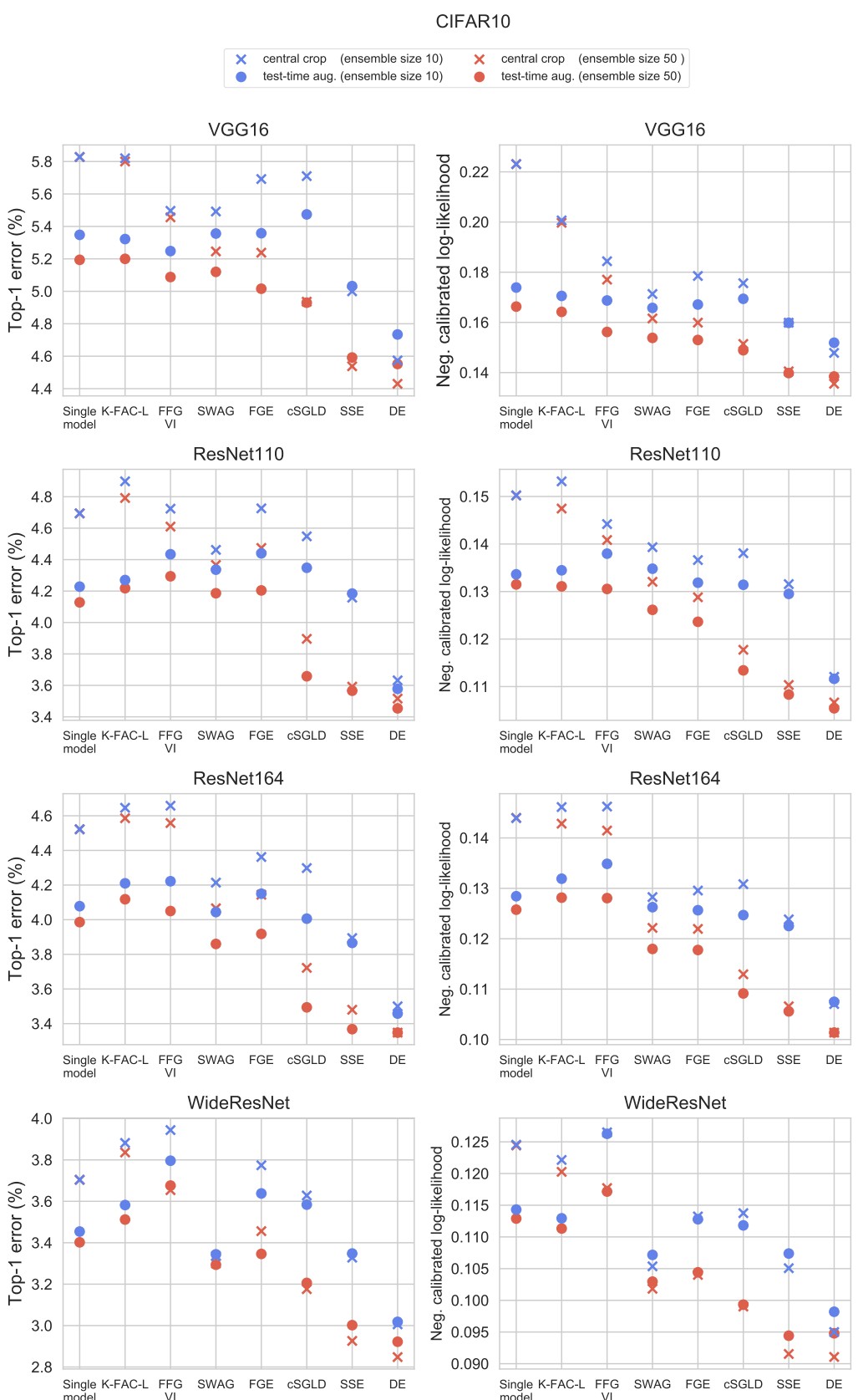

Figure 14: Classification error and negative calibrated log-likelihood before vs. after test-time augmentation on CIFAR-10

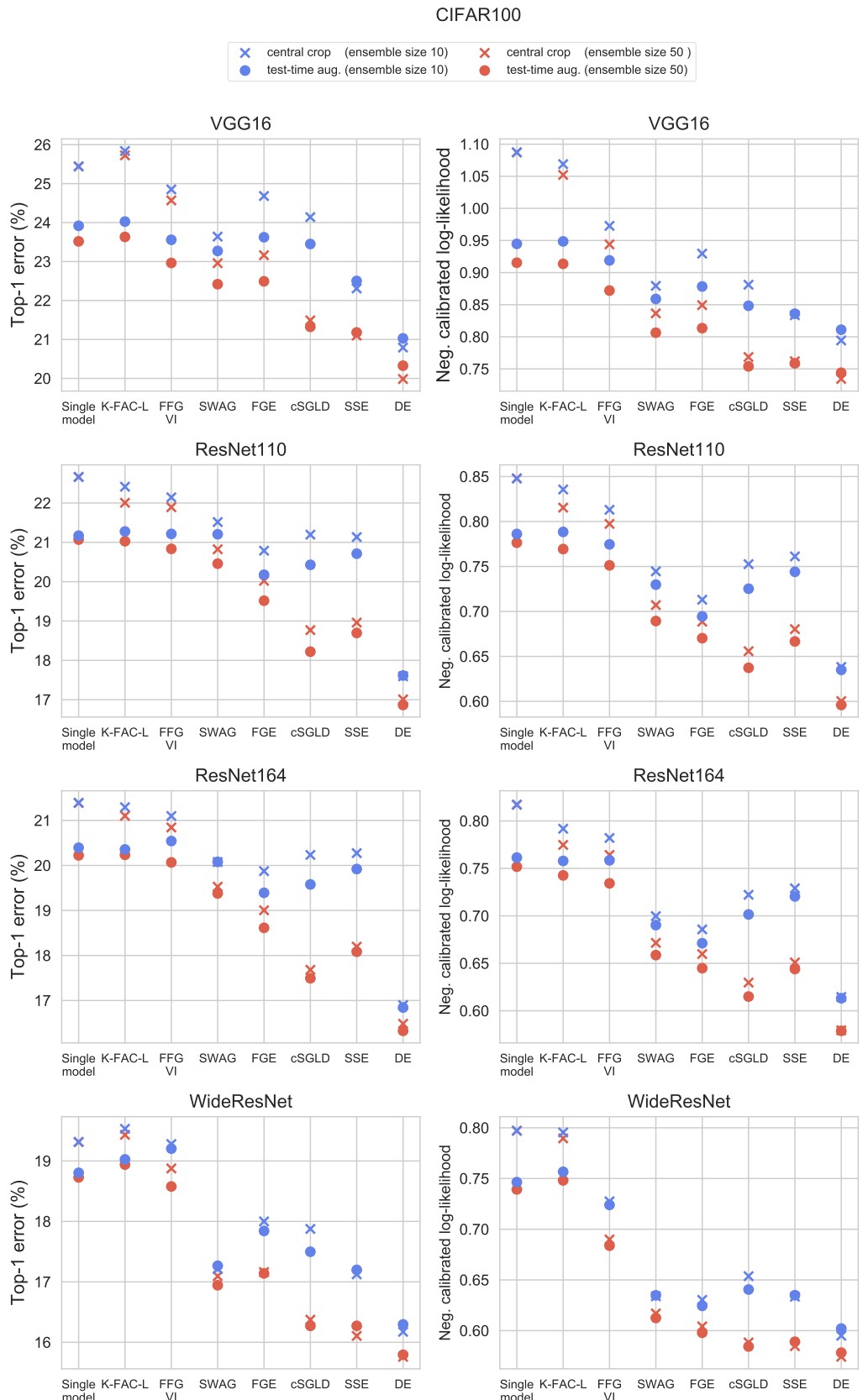

Figure 15: Classification error and negative calibrated log-likelihood before vs. after test-time augmentation on CIFAR-100

