# OpenReview forum: "Pitfalls of In-Domain Uncertainty Estimation and Ensembling in Deep Learning"
_ICLR.cc/2020/Conference — Accept (Poster)_

### Official Review · AnonReviewer2 · 2019-10-22
**Official Blind Review #2**

**Rating:** 8

**Review:**

The paper provide an extensive review of current advances in uncertainty estimation in neural networks with the analysis of drawbacks of currently used uncertainty metrics and comparison on scale the recent method to estimate uncertainty. The paper covers a lot of uncertainty metrics and a wide range of methods. The paper focuses on in-domain uncertainty estimation complementing the recent similar review on out-of-domain uncertainty estimation.

It seems that the paper provides the analysis missing in the current literature. Whereas as mentioned Yukun Ding in a public comment there is a related work on identifying issues with popular uncertainty metrics, the mentioned paper is missing the through comparison of the methods for estimating uncertainty.

Such kind of thorough analysis (especially performed on scale on large datasets) and comparison is of obvious interest to the community as well as objective comparison of the current state-of-the-art.

The paper is clearly written and easy to follow.

Based on this, I believe this is a strong technical paper and it should be accepted. However, the analysis in the paper is not overwhelmingly exhaustive. Some of the arguments on that are listed below.

Below is the list of comments/thoughts for potential improvement of the paper:
1.	“In this case, a model is expected to provide correct probability estimates:” – may be not the best choice of words, because for out of domain uncertainty estimation we still expect a model to provide correct probability estimates
2.	The first paragraph on page 3 seems to better fit in Section 2, for example, on the very beginning of Section 2.
3.	“Comparison of the log-likelihood should only be performed at the optimal temperature.” and others alike – personally, I do not support this kind of formatting for a scientific paper
4.	“can produce an arbitrary ranking of different methods. <…> Empirically,” – in the current form it seems that the first statement is somehow theoretically justified and then additionally it is confirmed empirically in this paper. I believe that the authors use empirical observation itself as the justification of the first statement, if that the case it should be reworded here. For example, “can produce an arbitrary ranking of different methods as we show below/ as we show empirically. We demonstrate that the overall …” If my belief is incorrect and there are other grounds that justify the first statement that it is required a reference after this statement.
5.	Italic and non-italic LL usage is unclear
6.	Having “Brier score” emphasised as a paragraph, it seems that there should be a paragraph log-likelihood as well
7.	“In that case, both objects and targets of induced binary classification problems remain fixed for all models” – do the authors consider in this case all out-of-domain objects as having a positive class and all in-domain objects as having a negative class? Because the models are still going to make individual misclassification mistakes
8.	Figure 2 – legend occupies too much space of the plot occluding almost a third part of the plot. Maybe taking the legend out of the plot to the right and squeezing the plot to make a room for the legend would be a better solution
9.	In eq. (4) and (5) subscript DE is not defined
10.	“SSE and cSGLD outperform all other techniques except deep ensembles” –cSGLD was not applied on ImageNet, therefore this statement is a bit misleading
11.	Colour of SWAG in Figure 3 is not very clear. Only excluding other colours I can determine which line is SWAG. Similar to cSGLD, is seems that SWAG was not applied on ImageNet. Why is that if that is the case? And it should be clearly stated at least in experimental setup in Supplementary
For colours in general, lines in legends are very thin and it is difficult to assess their colour. I appreciate the authors compare a lot of methods and therefore have to use a lot of colours, but it is quite difficult to assess them even on screen not to mention if the paper is printed out. Could the authors please use thicker lines in legends at least?
12.	“Being more “local” methods” – without any context in the main paper this referral to “local” methods is unclear. Also it is good to add a reference to Appendix review of the considered methods in the main text.
13.	Missing details of what kind of augmentation is used in Section 4.3. Is it the same as training augmentation specified in Supplementary? It would require a reference to Supplementary
14.	“(Figure 1, Table REF)” – missing number for Table
15.	“Our experiments demonstrate that ensembles may be severely miscalibrated by default while still providing superior predictive performance after calibration.” – unclear which experiments demonstrate this and superior in comparison to what
16.	The issues of uncalibrated log-likelihood and TACE are clearly shown in the paper, whereas the issues with misclassification detection are only verbally discussed. An illustrative example, at least a toy thought example, could really improve the paper here
17.	The chosen main performance metric is not very convincingly motivated. It is clear why it is based on calibrated log-likelihood, but it is not very convincing why one cannot just used calibrated log-likelihood as a performance metric, why one should base the metric on deep ensembles instead. Also from the long-term perspective, if the community comes up with methods clearly outperforming deep ensembles, the metric would need to be based on one of these new methods
18.	There is an indirect uncertainty metric that is not mentioned in the paper – uncertainty used in active learning (see, e.g., Hernández-Lobato and Adams, 2015. Probabilistic backpropagation for scalable learning of Bayesian neural networks)
19.	Figures 4 and 5 are too small
20.	“the original PyTorch implementations of SWA” – SWA is not considered in the paper
21.	“hidden inside an optimizer … The actual underlying optimization problem” – it seems that the ICLR audience should be familiar with “actual optimization problems” rather than using blindly the optimizer. It is always good to explicitly write down an equation that is used in a paper, but this wording seems a bit off for ICLR
22.	“\hat{p}(y^∗_i = j | x_i, w) denotes the probability that a neural network with parameters w assigns to class j when evaluated on object x_i” – it should be \hat{p}(y_i = j | x_i, w), y^*_i is observed
23.
a.	Why was dropout applied only for limited number of architectures and not applied on ImageNet at all?
b.	Why wasn’t cSGLD applied on ImageNet
24.	“On CIFAR-10/100 parameters from the original paper are reused” – it is better to repeat the reference here


Minor:
1.	Font size in eq. (10) should be the same as the rest of the paper
2.	“Or models achived top-1 error of”: “Or” - ?, “achived” -> achieved
3.	“for a 45 epoch form a per-trained model”: “form” -> “from”?

**Experience Assessment:**

I have published one or two papers in this area.

**Review Assessment: Checking Correctness Of Derivations And Theory:**

N/A

**Review Assessment: Checking Correctness Of Experiments:**

I carefully checked the experiments.

**Review Assessment: Thoroughness In Paper Reading:**

I read the paper at least twice and used my best judgement in assessing the paper.

---

> ### Author Response · Authors · 2019-11-13
> **Response to Review #2 (part 1/2)**
>
> We would like to sincerely thank you for your thoughtful and thorough remarks. They will allow us to significantly improve the quality of our paper in its next revision.
>
> We address the major questions below:
>
> > 7.	“In that case, both objects and targets of induced binary classification problems remain fixed for all models” – do the authors consider in this case all out-of-domain objects as having a positive class and all in-domain objects as having a negative class? Because the models are still going to make individual misclassification mistakes
>
> Yes, in the case of out-of-domain detection the out-of-domain objects have the positive class and the in-domain objects have the negative class. Because of that, both the objects and the labels of the auxiliary binary classification problems of out-of-domain detection remain the same across different models.
>
> > 9.	In eq. (4) and (5) subscript DE is not defined
>
> We define CLL_m where m stands for the name of the ensembling technique. DE stands for deep ensemble. We will clarify this in the future revision of the paper.
>
> > 11.	… Similar to cSGLD, is seems that SWAG was not applied on ImageNet. Why is that if that is the case? And it should be clearly stated at least in experimental setup in Supplementary. …
> > 23.	… Why was dropout applied only for limited number of architectures and not applied on ImageNet at all? ...
>
> Due to limited computational resources we have prioritised SSE over cSGLD and FGE over SWAG since they are closely related to each other and achieve similar performance on CIFAR10/100 datasets. Since these techniques are quite similar to each other, we expect these results to translate well to ImageNet. Also, we have only applied dropout to the VGG model and the WideResNet model since plain ResNets are conventionally trained without dropout, and naive application of dropout hurts the final predictive performance.
>
> > 13.	Missing details of what kind of augmentation is used in Section 4.3. Is it the same as training augmentation specified in Supplementary? It would require a reference to Supplementary
>
> For test-time augmentation we use the same data augmentation as used during training. We will add the reference to Supplementary there.
>
> > 15.	“Our experiments demonstrate that ensembles may be severely miscalibrated by default while still providing superior predictive performance after calibration.” – unclear which experiments demonstrate this and superior in comparison to what
>
> The most drastic difference can be observed in Figure 1. Deep ensembles + augmentation (DE+aug) on ImageNet has exteremely poor calibration and perform worse than plain DE, whereas calibrated DE+aug outperforms plain DE (and other techniques as well).

---

> ### Author Response · Authors · 2019-11-13
> **Response to Review #2 (part 2/2)**
>
> > 16.	The issues of uncalibrated log-likelihood and TACE are clearly shown in the paper, whereas the issues with misclassification detection are only verbally discussed. An illustrative example, at least a toy thought example, could really improve the paper here
>
> AUROC/AUPR for misclassification detection plainly provides numbers that can not be compared across different models. We will try to come up with a convincing illustrative example, but it is not yet clear for us how to make it more convincing than the verbal discussion.
>
> > 17.	The chosen main performance metric is not very convincingly motivated. It is clear why it is based on calibrated log-likelihood, but it is not very convincing why one cannot just used calibrated log-likelihood as a performance metric, why one should base the metric on deep ensembles instead. Also from the long-term perspective, if the community comes up with methods clearly outperforming deep ensembles, the metric would need to be based on one of these new methods
>
> DEE is basically a more convenient way to visualize the calibrated log-likelihood. The calibrated log-likelihood does indeed seem to be a great absolute measure of performance. However, it is not very convenient if one wants to compare the performance of different ensembling techniques. Different models and datasets have different base values of calibrated log-likelihood, and its dependence on the number of samples is non-trivial. DEE is model- and dataset-agnostic and provides some useful insights that can be difficult to visualize using the calibrated log-likelihood alone.
>
> In the long run, the DEE metric might still be useful and insightful. While the DEE curve of deep ensembles is an identity function, superior methods would typically result in a higher DEE curve. We would only run into problems if this new superior method would outperform extremely large deep ensembles (say hundreds or thousands of samples) using just a handful of samples. However, we do not expect such a drastic gap to appear soon.
>
> > 18.	There is an indirect uncertainty metric that is not mentioned in the paper – uncertainty used in active learning (see, e.g., Hernández-Lobato and Adams, 2015. Probabilistic backpropagation for scalable learning of Bayesian neural networks)
>
> We do mention active learning in the related works section, but this reference is indeed very relevant here. Thank you.
>
> > 20.	“the original PyTorch implementations of SWA” – SWA is not considered in the paper
>
> While we do not use SWA in our experiments, our codebase is heavily based on the original implementation of SWA since it allowed to easily reproduce the training of different models and was easy to modify for our needs. We will articulate the reference more clearly in the next revision of the paper.

---

### Official Review · AnonReviewer1 · 2019-10-23
**Official Blind Review #1**

**Rating:** 6

**Review:**

The authors response on 13th nov regarding the main concerns I have are valid. They make sense. I thank the authors for the detailed explanations. I went back and did another thorough read of the work. As it stands, I am OK to change my review to weak accept.

---------------------------------

The authors evaluate a variety of ensemble models in terms of their ability to capture in-domain uncertainity. A set of metrics are used to perform these evaluations and in turn, using deep ensenble as a reference, the authors study the behaviour/capacity of the rest of the methods.

Although the motivation for the work is sensible, there are several critical issues with the paper and the summary is not necessarily conclusive in terms of gaining any new insights.
1. Much of the evaluations rely on the choice/nature of the optimal temperature, which would be different for different models. The authors suggest to use the model-specific optimal values when comparing instead of fixing the temperature? Why is this the case? Further, if we take this into account (i.e., allow for comparing different temperatures) then much of the differences between DEE and others cannot be directly interpreted. This is the case when using log-likelihood and Brier scores.
2. AUC can be transformed into a normalized probability distribution (CDF), and hence in principle it is model/hyperparameters agnostic. This is one of the reasons i is used as information criterion in Bayesian model selection. Area of AUC is a valid metric as well. To that end, why do the authors suggest that it cannot be used as criteria for comparison across models?
3. From section 3.5 it is not clear how test time cross validation is tackling temperature scaling?
4. In section 4.1, the hypothesis on #independent trained networks is great and it makes sense? How is this translating ito the evaluations? None of the results actually talk about this aspect directly? Or am I missing something here?
5. Setting the evaluations with DEE as reference is problematic because we already know from random sampling theory that deep ensemble is better than the normal ensembles (tech results on random sampling for model fitting and RL etc. optimization results on mode finding with single mode vs. multi model methods also say similar things) and in fact that was the main motivation. Also normal regularization (like dropout or K-facL are more towards overfitting than ensembling) are not really an ensemble. Putting these together, most of the conclusions and the lots (figure 3 in particular) is by definition true. Nothing surprising.



**Experience Assessment:**

I have published one or two papers in this area.

**Review Assessment: Checking Correctness Of Derivations And Theory:**

I carefully checked the derivations and theory.

**Review Assessment: Checking Correctness Of Experiments:**

I carefully checked the experiments.

**Review Assessment: Thoroughness In Paper Reading:**

I read the paper thoroughly.

---

> ### Author Response · Authors · 2019-11-13
> **Response to Review #1 (part 1/2)**
>
> We would like to sincerely thank you for your thoughtful remarks and questions.
>
> We address your concerns below:
>
> > 1. Much of the evaluations rely on the choice/nature of the optimal temperature, which would be different for different models. The authors suggest to use the model-specific optimal values when comparing instead of fixing the temperature? Why is this the case? Further, if we take this into account (i.e., allow for comparing different temperatures) then much of the differences between DEE and others cannot be directly interpreted. This is the case when using log-likelihood and Brier scores.
>
> Most of the modern deep learning techniques are overconfident in their predictions, i.e. the effective temperature is lower than optimal. Moreover, it does not seem possible for now to determine the optimal temperature relying only on the training data. Validation-based temperature scaling is a simple yet powerful calibration technique that allows to improve many predictive performance metrics, e.g. the test log-likelihood, post-hoc for all methods and models. We would like to reduce the influence of the non-optimal temperature on the predictive performance, so we see the temperature scaling as an essential step in training the model. As we show in Figure 1, comparing different techniques without temperature scaling can yield misleading results. E.g., deep ensembles with test-time data augmentation (DE+augment) seem to perform worse than deep ensembles without data augmentation (DE) in terms of the log-likelihood, whereas after temperature scaling DE+augment outperforms plain DE.
>
> Comparing methods at different temperatures is fair since the procedure for temperature scaling is the same for all methods, as described in Section 3.5. Moreover, we stress that this setting is more reasonable compared to using the same temperature for all methods in the benchmark since different methods have different optimal temperatures on hold-out data.
>
> > 2. AUC can be transformed into a normalized probability distribution (CDF), and hence in principle it is model/hyperparameters agnostic. This is one of the reasons i is used as information criterion in Bayesian model selection. Area of AUC is a valid metric as well. To that end, why do the authors suggest that it cannot be used as criteria for comparison across models?
>
> Metrics like AUROC / AUPR cannot be used for a particular problem of misclassification detection. Let us summarize the argument. We aim to compare different models---trained on the same data---in terms of ability to distinguish between correct and wrong classifications. The prior literature suggests the following: i) every prediction of a particular model receives a confidence score, ii) the score then is treated as an output of a binary classifier that detects misclassifications. AUROC / AUPR of these binary classifiers is used for comparison between different models.
>
> Such a comparison, however, is not correct. Every model has its own correct and wrong predictions, and thus poses its own misclassification detection problem (binary classification of correctly classified vs. incorrectly classified examples). Particularly, in the case of comparison between K models, we have K different misclassification detection datasets comprised of pairs (original object, “correctly classified” / “incorrectly classified” binary label) with a different labeling for each model. The described comparison procedure essentially corresponds to a comparison of performance of classifiers that solve different classification problems. Such metrics are incomparable.
>
> > 3. From section 3.5 it is not clear how test time cross validation is tackling temperature scaling?
>
> The “test-time cross-validation” method for evaluating metrics at an optimal temperature is organized as follows:
> 1. A test set is randomly shuffled and divided into K folds of the same size
> 2. A temperature T* is adjusted by K-1 folds, T* = argmax_T LL(Model(T), Data(K-1 folds))
> 3. The model at the optimal temperature T* is used to evaluate metrics on the Kth fold.
> 4. The steps 1-3 are repeated several times, the metric values are averaged.
> In our experiments we use K=2.
>
> In the step 2 we solve a 1D optimization problem that optimizes LL on K-1 folds w.r.t. a scalar temperature T. The result of step 2 may differ depending on the particular data split. Strictly speaking, the described algorithm estimates expectation of test metrics w.r.t. the distribution of optimal temperatures induced by the random data splits. In practice, we noticed that the optimal temperatures did not differ much on different splits.

---

> ### Author Response · Authors · 2019-11-13
> **Response to Review #1 (part 2/2)**
>
> > 4. In section 4.1, the hypothesis on #independent trained networks is great and it makes sense? How is this translating ito the evaluations? None of the results actually talk about this aspect directly? Or am I missing something here?
>
> This point presumably refers to the following sentence in section 4.1: “What number of independently trained networks yields the same performance as a particular ensembling method?” This is not intended to be a presentation of hypothesis. This question only sets the stage for the introduction of the deep ensemble equivalent (DEE) metric which directly answers the question when evaluated. DEE plays a major role in our benchmark and we have quantitative results concerning it (e.g. Figure 3 in the main text). We discuss the results related to DEE in Section 4.2 and Section 5 and provide more detailed results in Appendix E.
>
> On the other hand, if the comment refers to the following sentence "Deep ensembles, ..., which can intuitively result in a better ensemble.", it was just a motivation to consider deep ensembles as a potentially strong baseline.
>
> > 5. Setting the evaluations with DEE as reference is problematic because we already know from random sampling theory that deep ensemble is better than the normal ensembles (tech results on random sampling for model fitting and RL etc. optimization results on mode finding with single mode vs. multi model methods also say similar things) and in fact that was the main motivation. Also normal regularization (like dropout or K-facL are more towards overfitting than ensembling) are not really an ensemble. Putting these together, most of the conclusions and the lots (figure 3 in particular) is by definition true. Nothing surprising.
>
> Did you mean DE (deep ensembles) instead of the DEE score here? The superior performance of deep ensembles is indeed not surprising. Highlighting this fact is not the main purpose of this paper. Instead, our study is largely aimed at comparing ensembling methods in a fair and interpretable way to gain insights in the fields of ensembling and uncertainty estimation.
>
> Methods that are based on stochastic computation graphs, e.g., MC-dropout, K-FAC Laplace and variational inference, are commonly regarded as ensembling techniques in the bayesian deep learning literature and are frequently used as a baseline in ensembling-based uncertainty estimation research (Gal and Ghahramani 2016, Lakshminarayanan et al 2017, Louizos and Welling 2017, Maddox et al 2019). Deep ensembles, in contrast to dropout, are rarely considered as a baseline in the bayesian deep learning literature, but we believe that overcoming the DE baseline is a strong challenge for the community.
>
> We would highly appreciate it if you could provide links for the papers mentioned in your review since they seem to be relevant to our study.
>
> Louizos C, Welling M. Multiplicative normalizing flows for variational bayesian neural networks, ICML 2017.
> Lakshminarayanan B, Pritzel A, Blundell C. Simple and scalable predictive uncertainty estimation using deep ensembles, NeurIPS 2017.
> Maddox W, Garipov T, Izmailov P, Vetrov D, Wilson AG. A simple baseline for bayesian uncertainty in deep learning, NeurIPS 2019.
> Gal Y, Ghahramani Z. Dropout as a bayesian approximation: Representing model uncertainty in deep learning, ICML 2016.

---

### Official Review · AnonReviewer3 · 2019-10-24
**Official Blind Review #3**

**Rating:** 6

**Review:**

Summary:
This paper mainly concerns the quality of in-domain uncertainty for image classification. After exploring common standards for uncertainty quantification, the authors point out pitfalls of existing metrics by investigating different ensembling techniques and introduce a novel metric called deep ensemble equivalent (DEE) that essentially measures the number of independent models in an ensemble of DNNs. Based on the DEE score, a detailed evaluation of modern DNN ensembles is performed on CIFAR-10/100 and ImageNet datasets.

Strengths:
The paper is well written and easy to follow. The relationship to previous works is also well described. Overall, I think this is a good paper, which gives a detailed overview of existing metrics for accessing the quality in in-domain uncertainty estimation. The idea behind the proposed DEE score is nice and simple, clearly showing the quality of different ensembling methods (in Fig. 3). Given the importance of uncertainty analysis to deep learning, I believe this work will have a positive impact on the community.

Weaknesses:
- On page 6, the authors mention that the prior in Eq. 3 is taken to be a Gaussian N(\mu, diag(\sigma^2)) for Bayesian neural networks, however, many other choices of a prior distribution are available in the literature. What is the impact of changing prior distributions on the quality of uncertainty estimates in the case of variational inference?
- Data augmentation is commonly used for improving model performance. However, I find the results presented in Sect 4.3 are not clear enough, note that for a given ensembling method in Table 1, the negative calibrated log-likelihood may increase or decrease when using different networks (VGG, ResNet, etc.). I think it would be interesting to elaborate a bit more on the influence of model complexity.
- On page 15, in Eq. 12, the choice of the variance parameter \sigma_p^2=1/(N*wd) seems unclear and should be better explained.

Minor comments:
The size of some figures appears too small, for example Fig. 4 and Fig. 5, which may hinder readability.

**Experience Assessment:**

I have read many papers in this area.

**Review Assessment: Checking Correctness Of Derivations And Theory:**

I assessed the sensibility of the derivations and theory.

**Review Assessment: Checking Correctness Of Experiments:**

I assessed the sensibility of the experiments.

**Review Assessment: Thoroughness In Paper Reading:**

I read the paper at least twice and used my best judgement in assessing the paper.

---

> ### Author Response · Authors · 2019-11-13
> **Response to Review #3**
>
> We would like to sincerely thank you for your thoughtful remarks and questions.
>
> We address your concerns below:
>
> > On page 6, the authors mention that the prior in Eq. 3 is taken to be a Gaussian N(\mu, diag(\sigma^2)) for Bayesian neural networks, however, many other choices of a prior distribution are available in the literature. What is the impact of changing prior distributions on the quality of uncertainty estimates in the case of variational inference?
>
> The prior distribution is a part of the underlying probabilistic model, whereas most ensembling techniques can be considered as approximate inference techniques under such a model. Aiming for a fair comparison, we set the probabilistic model (and, therefore, the prior distribution) to be the same across all ensembling techniques. We use the Gaussian prior, induced by the optimizer favored in recent literature since it is simple and provides reasonably high performance. However, it would indeed be interesting to see how the choice of the prior influences different ensembling techniques.
>
> > Data augmentation is commonly used for improving model performance. However, I find the results presented in Sect 4.3 are not clear enough, note that for a given ensembling method in Table 1, the negative calibrated log-likelihood may increase or decrease when using different networks (VGG, ResNet, etc.). I think it would be interesting to elaborate a bit more on the influence of model complexity.
>
> The effect appears due to std of the negative calibrated log-likelihood (nCLL). In all the cases where nCLL may increases or decreases within one method the difference has the order of ~1e-3 or less and lies within std interval. We will correct the tables and add stds.
>
> On CIFAR datasets test-time data augmentation helps “weak” ensembling methods like dropout, K-FAC Laplace and variational inference, whereas on ImageNet we observe the improvement for all techniques. We hypothesize that this is caused by a more diverse data augmentation on ImageNet as compared to CIFAR. We will move the ImageNet results (Table 9) into the main section of the paper as they seem to be more representative than CIFAR results. It would be interesting to see whether the use of more diverse data augmentation (e.g. rotations, color transformation, etc.) improves stronger ensembles as well.
>
> > On page 15, in Eq. 12, the choice of the variance parameter \sigma_p^2=1/(N*wd) seems unclear and should be better explained.
>
> It is the same prior as defined in eq. 9 on page 14. The weight decay parameter wd is the coefficient before the L2 regularizer in the objective. In most deep learning frameworks, one computes the *average* loss in the minibatch instead of the *sum* across all objects. Therefore, one needs to rescale this coefficient by the size of the training set to obtain the underlying prior distribution. We will update the paper with a more clear explanation.

---

### Public Comment · ~Yukun_Ding1 · 2019-10-08
**A related work**

Thanks for the great work. There is a related work that has a similar finding on the pitfalls of existing metrics. You might want to cite it in related works. Thanks.
https://arxiv.org/abs/1903.02050

---

> ### Author Response · Authors · 2019-11-13
> **Response to "A related work"**
>
> Thank you. This work appears relevant and we will cite it in the next revision of our paper.

---

### Author Response · Authors · 2020-04-28
**Additional links**

Additional links:

- Blog: https://senya-ashukha.github.io/pitfalls-uncertainty&ensembling

---

### Decision · Program_Chairs · 2019-12-19

**Decision:**

Accept (Poster)

**Comment:**

The paper points out pitfalls of existing metrics for in-domain uncertainty quantification, and also studies different strategies for ensembling techniques.

The authors also satisfactorily addressed the reviewers' questions during the rebuttal phase. In the end, all the reviewers agreed that this is a valuable contribution and paper deserves to be accepted.

Nice work!